# Optical neural network via loose neuron array and functional learning

Yuchi Huo[1,2,3], Hujun Bao [1,2] ✉, Yifan Peng[4], Chen Gao[2], Wei Hua[2], Qing Yang[2], Haifeng Li[2], Rui Wang[1] ✉ & Sung-Eui Yoon[3] ✉

This research proposes a deep-learning paradigm, termed functional learning (FL), to physically train a loose neuron array, a group of non-handcrafted, non-differentiable, and loosely connected physical neurons whose connections and gradients are beyond explicit expression. The paradigm targets training non-differentiable hardware, and therefore solves many interdisciplinary challenges at once: the precise modeling and control of high-dimensional systems, the on-site calibration of multimodal hardware imperfectness, and the end-to-end training of non-differentiable and modeless physical neurons through implicit gradient propagation. It offers a methodology to build hardware without handcrafted design, strict fabrication, and precise assembling, thus forging paths for hardware design, chip manufacturing, physical neuron training, and system control. In addition, the functional learning paradigm is numerically and physically verified with an original light field neural network (LFNN). It realizes a programmable incoherent optical neural network, a well-known challenge that delivers light-speed, high-bandwidth, and power-efficient neural network inference via processing parallel visible light signals in the free space. As a promising supplement to existing power- and bandwidth-constrained digital neural networks, light field neural network has various potential applications: brain-inspired optical computation, high-bandwidth power-efficient neural network inference, and light-speed programmable lens/displays/detectors that operate in visible light.

Deep learning has recently experienced tremendous success in delivering impressive results in various domains, including computer vision, medical analysis, natural language processing, and automatic vehicles[1,2]. Its primary process is to train multiple layers of artificial neural networks to find abstractions in domain-specific datasets. The universal approximation theorem states that a feed-forward network with a single hidden layer containing a finite number of neurons can approximate continuous functions on compact subsets of $\mathbf{R}^n$ under mild assumptions pertaining to the activation function[3].

Modern electronic devices offer strong computing capabilities for the processing of laborious matrix multiplications and memory references, leading to the development of advanced networks with ever-growing amounts of neuron connections. However, the stringent constraints on power and bandwidth hinder the broader applications of deep learning in mobile devices, embedded systems, wireless wearable devices, and robots. The optical neural network (ONN) is a promising alternative mechanism for high-bandwidth, low-latency, and power-efficient neural network inference[4–7]. However, building a programmable ONN using electronically controlled optical components remains an open problem. Note that while electronically controlled optical components are commonly modeled as perfect surfaces, they actually have inconsistently unmeasurable microstructures and materials, creating unpredictable inter-reflections and interference. In addition to the optics, the electronic circuit and

[1]State Key Lab of CAD&CG, Zhejiang University, Hangzhou, China. [2]Zhejiang Lab, Hangzhou, China. [3]Korea Advanced Institute of Science and Technology, Deajeon, South Korea. [4]The University of Hong Kong, Hong Kong, SAR, China. ✉e-mail: bao@cad.zju.edu.cn; ruiwang@zju.edu.cn; sungeui@gmail.com

physical environment also introduce unpredictable bias and variance. As such, existing approaches can no longer properly simulate or train programmable ONNs.

In this work, we conceive the concept of practically training loose neuron arrays for practical tasks. It is a physical counterpart of the artificial neural network and enables the transformation from hand-crafted hardware designs to non-handcrafted designs. One of the most important recent discoveries is that merely connecting many artificial neurons in a non-handcrafted way outperforms sophisticated hand-crafted algorithms in a large number of tasks. While connecting arti-ficial neurons is very handy in computers, there is no trivial way to connect and train arbitrary physical neurons because the actual design, material, structure, manufacture, fabrication, and run-time environment are far from ideal models. On the other hand, enabling the connection and training of arbitrary physical neurons releases us from such constraints. It reveals possibilities for hardware design, chip manufacturing, and system control.

This concept is instantiated for the realization of incoherent pro-grammable ONNs. We numerically verify the feasibility of various neuron arrays with liquid-crystal (LC) hardware to form programmable ONNs (Fig. 1). The neuron arrays are stacks of electronically controllable polarization rotators trained to physically modulate incoherent light signals for various inference tasks as a fully connected neural network. The interconnectivity between two adjacent neural network layers is inherently executed by light propagation between input and output neuron pairs, one input neuron on the input plane as wave source, and the output neuron on the following output plane. The weights of con-nections are modulated by an electronically controlled polarization field, followed by polarizing filters. The LC neurons, i.e., the hardware para-meter of the LCs, are trained by a functional learning (FL) paradigm.

We physically implement a regular-2 array with off-the-shelf opti-cal components, e.g., liquid-crystal display (LCD) panels, polarizers,

and a camera, without fine calibration. We call such a programmable incoherent ONN design a light field neural network (LFNN, Fig. 2). At runtime, the LFNN can perform a specific inference task for which the parameters are trained. One engineered LFNN can shift to different inference tasks at runtime by simply switching to different parameters and can function as a power-efficient light-speed compute unit or as programmable lens/displays.

To the best of our knowledge, there is no trivial way to train an arbitrary neuron array, e.g., the LFNN. Classic computational optics abstracts a real-world optical system with a highly simplified analytical model to track light propagation. However, real-world systems have much more complicated micro-structures and interactive effects among elementary components such as electrodes, glass substrates, color filters, thin-film transistors, or liquid crystals, making the neurons non-differentiable. Moreover, the optics undergo mutual interference with other spectra, electricity, temperature, and moisture. In practice, an analytical model can hardly simulate with high accuracy and train such a high-dimensional real-world system that contains many non-differentiable parameters. Importantly, our prototype is built upon inexpensive optical components, which feature high noise, low light transmittance, narrow viewing angles, poor linearity, neuron incon-sistency, and biased alignments.

Instead, we propose the FL paradigm to control such an uncali-brated optical system. Given the input signals and expected targets as the training set, the parameters of the system are trained as trainable neurons to undertake inference on the input signal:

$$\min_{p} \quad \|f(x; p) - y\|, \tag{1}$$

where $x$ is the normalized input signal, $y$ is the target output, $p$ is the primary parameter of the system, $\|\cdot\|$ denotes a general distance metric, and $f(\cdot; p) \in \mathcal{C} : [0, 1]^{|x|} \to [0, 1]^{|y|}$ is an unknown function in the

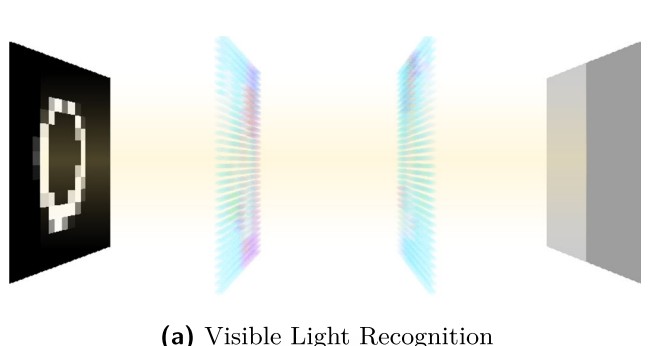

**(a)** Visible Light Recognition

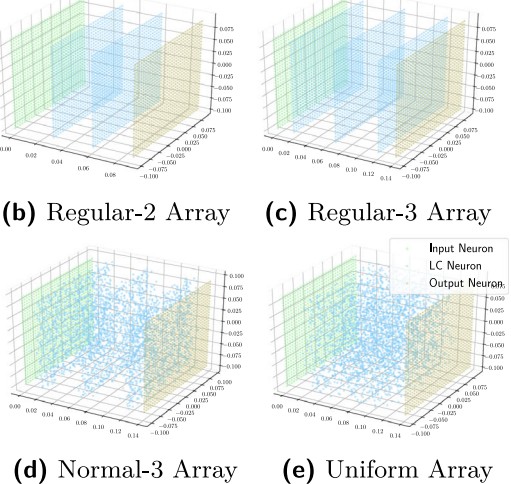

**(b)** Regular-2 Array    **(c)** Regular-3 Array

**(d)** Normal-3 Array    **(e)** Uniform Array

|  | Regular-2 | Regular-3 | Normal-3 | Uniform |
|---|---|---|---|---|
| **LC Neurons** | 2048×3 | 3072×3 | 3072×3 | 3072×3 |
| **1-Layer MNIST** | 91.03% | 92.07% | 92.40% | 92.45% |
| **2-Layer MNIST** | 96.61% | 97.30% | 97.55% | 97.65% |
| **3-Layer CIFAR10** | 47.48% | 50.61% | 51.73% | 52.53% |

**(f)** Accuracy of Losse Neural Arrays

**Fig. 1 | Loose neuron array.** We numerically verify loose neuron arrays with simulation. **a** The input neurons are point light sources, each with a 70 field of view, output neurons are energy gatherers, and LC neurons can attenuate incoherent light. These arrays are treated as black-box systems without gradient models and can be successfully trained using our FL paradigms. **b** Regular-2 array has two LC panels of a total of 2048 regularly aligned LC neurons. **c** Regular-3 array contains three layers of a total of 3096 LC neurons. **d** Normal-3 array is built on (**b**) with random jitters. **e** Uniform array has 3096 randomly distributed LC neurons. **f** The uniform array with random LC unions gains the highest performance, showing that data-driven training paradigms can outperform hand-craft hardware designs.

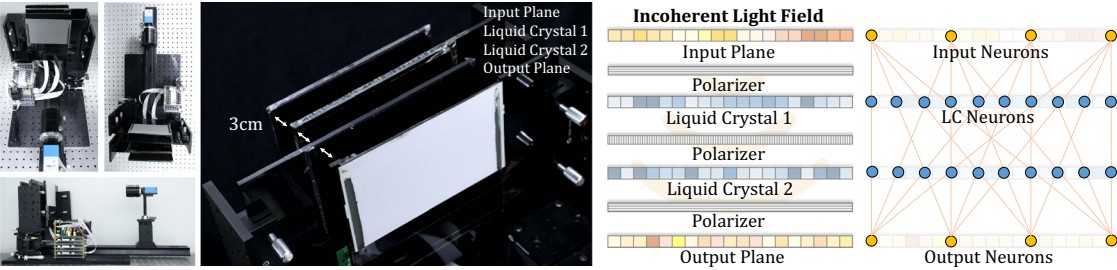

**Fig. 2 | Light field neural network (LFNN).** The LFNN prototype consists of an input plane, an output plane, two liquid-crystal layers, and three perpendicular linear polarizers. The output plane is a scattering plane followed by a camera to acquire the data. We use an LCD as the input plane by representing artificial neurons with pixels. Like a fully connected neural network layer, these neurons of the input plane synergistically activate the artificial neurons of the next layer through light propagation. We adopt two LCDs as controllable liquid-crystal components by eliminating their backlights and polarizing films. The key idea of the LFNN is to train the pixel control parameters of the liquid-crystal components to distort the polarization directions of the input light so as to synthesize the attenuated light paths as a weighted connection between the input and output planes. In addition to the parameters of LCs, we modify the driver so that the neuron-wise intensity gains of the input and output planes are trainable parameters to increase the degree of freedom. The resolutions of the input, output, and LCs are all $32 \times 32$ with RGB channels in our prototype.

functional space mapping the continuous input $x$ to the target $y$. In our specific problem, $f(\cdot;p)$ is the optics of the LFNN hardware, $p$ is the hardware control parameters, $x$ is the input, and $y$ is the expected output. At a specific inference task, once $p$ is applied to the hardware, the incoherent light signal passing through the device is passively modulated to the target output.

Because $f(\cdot;p)$ has no explicit mathematical formulation, we introduce a functional neural network (FNN, Fig. 3a) to represent a learnable functional space and approximate Equation (3) as follows:

$$\min_{p,z} \quad \left\| f(x;) - g_N(z)(x;p) \right\| + \left\| g_N(z)(x;p) - y \right\|, \quad (2)$$

where $g_N(\cdot)(\cdot;\cdot)$ is an FNN, $z$ denotes the functional space coefficients, $p$ is the primary parameter as trainable neurons, and the two terms are iteratively optimized by $p-learning$ and $z-learning$ through the FL paradigm.

The FNN consists of a physically-inspired function basis block and the following nonlinear combination through five layers of convolutional neural network (CNN) with $3 \times 3 \times 64$ kernel and rectifier linear unit (ReLU)[2]. The function basis block reflects the physical structure of the device in which each input neuron has a potential impact, i.e., light propagation, to each output neural, forming a four-dimensional light field with numerous connections, whose cardinality is equal to input neural size times output neural sizes. The pixels of the LC panel, represented as control parameters, can potentially attenuate arbitrary input-to-output connection and are represented mathematically by multiplication to the input-to-output data flow. In order to depict features in various resolutions, we downsample the inputs and LC control parameters half and merge the results through trainable parameters. More details are in Supplementary document.

## Results
### Object classification
Object classification is an important benchmark for neural networks (Table 2a and Fig. 4). First, we use the MNIST dataset[8] to train and test a 1-layer LFNN. The input of the device is a $28 \times 28$ grayscale image, and the output is the predicted classification probability distribution of the input image, calculated by accumulating the sum of RGB channels at the output plane. Because the viewing angle of our backlight is too narrow to make the contribution of an input neuron cover the entire output plane, we train the probability distribution layout of different classes on the output plane to adapt to the uneven energy distribution. For example, the sum of the intensities in some dispersed areas represents the probability of one class (Fig. 4).

During the training, we iteratively conduct p-learning epochs and z-learning epochs until convergence. The p-data is the entire MNIST training set. The z-learning dynamically updates z-data at the beginning of each epoch by sending some impulses, i.e., 1024 randomly selected images from the MNIST training set, to the LFNN device and capturing its responses.

In the MNIST test set, the LFNN achieves a 91.02% prediction accuracy, and the FNN predicted accuracy is 91.39%. As a reference, one fully connected digital dense layer running in a computer achieves 92.71% accuracy. In comparison, existing training paradigms have a hard time training a large number of parameters of a high-dimensional system as the LFNN. Using the forward analytical model[9] with the back-propagation algorithm can train the parameters of the LCs, but calibrating the system is a challenging task. Treating the misalignment as a random variable of the training paradigm can alleviate the misalignment[10]. Similarly, we measure the actual physics, e.g., noise, misalignment, and narrow viewing angles, of the LFNN to build a forward model for the training. Specifically, we first measure the point spreading function (PSF) of every input neuron to the output plane. Then we measure and model the impact of every LC neuron on every input neuron's PSF as a linear function by switching on and off the corresponding neurons and recording the change on the output plane. In addition, we build look-up tables to calibrate the linearity of input and LC neurons by measuring the system output of the quantized control signal. Training the LFNN with the forward model achieves a 23.50% prediction accuracy. The limitation of this paradigm is that accurately measuring and modeling the tremendous amount of states of the LFNN, i.e., a high-dimensional, non-differentiable, non-linear, and correlated physical system, is beyond reality.

Alternative stochastic training paradigms like finite difference[6] and genetic algorithm[11] may work well for a small number of parameters, but are still hard to train a large number of parameters. The finite difference paradigm captures the difference quotient as gradients to optimize the parameters. While facing thousands of parameters, it has to measure the finite difference of every parameter, which requires capturing 6,144 output images from the LFNN for one input image. In contrast, our FNN paradigm captures 1024 output images to implicitly evaluate the gradients for 60,000 images at each epoch. As a result, using finite difference yields an 8.594% prediction accuracy within equal training time. Genetic algorithm faces a similar problem for large-scale optimization. While converging efficiently with a small number of parameters, stochastically exploring a high-dimensional space without gradients is futile. We leverage the

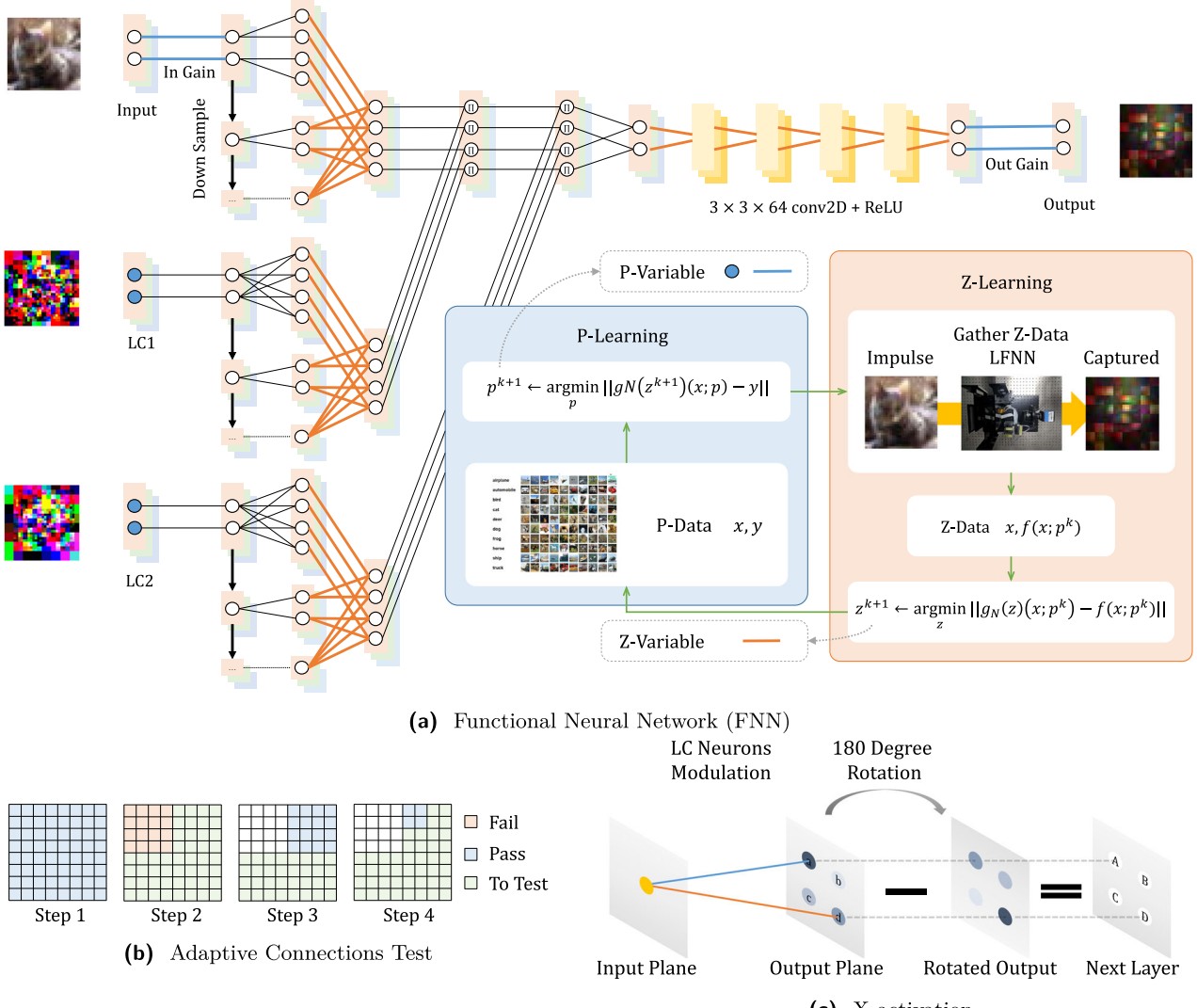

**(a)** Functional Neural Network (FNN)

**(b)** Adaptive Connections Test

**(c)** X-activation

**Fig. 3 | Functional learning (FL) paradigm.** This figure illustrates various details of the network design and training process. **a** The first part of the FNN is a physically-inspired functional basis block before the convolutional neural network (CNN) layers[32]. It reflects the physical structure of the device in which each input neuron has a potential impact, i.e., light propagation, to each output neuron, forming a four-dimensional light field with numerous connections, whose cardinality equals the input neural size times the output neural size. The pixels of the LC panel, represented as LC neurons, can potentially attenuate an arbitrary input-to-output connection and are represented mathematically by attenuation to the input-to-output data flow. In order to depict features in various resolutions, we downsample the input and LC neurons half and merge the results through trainable parameters.

The second part of the FNN consists of the subsequent five layers of the CNN with a $3 \times 3 \times 64$ kernel and the rectifier linear unit (ReLU)[2], which nonlinearly mixes the functional basis block. More details pertaining to the connections are given in supplementary document. **b** Illustration of the process of testing LC neuron connections by the quad-tree searching method. Illustrated here are the first four steps in a simple case. **c** By training the LC neurons' parameters, the input plane's neuron can either activate or deactivate a neuron in the next layer. For example, neuron a minuses neuron d equals neuron A. If the trained network produces a strong connection (marked with blue) and a weak connection (marked with orange), the yellow neuron activates neuron A and deactivates neuron D, and vice versa. The output of X-activation is the input of the next LFNN layer.

scikit-opt genetic algorithm to test training the LFNN. The metric of evaluating an offspring is calculated by randomly drawing 128 images and capturing the outputs. Within equal training time, the genetic algorithm trains the LFNN to achieve a 14.06% prediction accuracy. Table 1a summarizes the prediction accuracy using different training paradigms. Detailed discussion is in the supplementary document.

We also test multi-layer optical neural networks by cyclically feeding the captured outputs onto the input plane. This electronic cycling also conducts energy redistribution and nonlinear activation. We apply three neuron-wise activation operations. We refer to the first operation as X-activation, where the captured output is subtracted from its 180-degree rotated image. X-activation effectively enables consistent value distributions between layers, nonlinearities, and

deactivation, i.e., negative operations that are non-trivial in incoherent optical systems. Furthermore, its design uses only simple logic and operators that can be implemented by non-digital hardware in the future to match the activation to speed-of-light processing. The second operation is common neuron-wise batch normalization[12], followed by the third operation, a simple amplitude filter integrated into the driver of the input plane that clamps the impulse between zero and one. We connected multiple FNNs to train the multi-layer LFNN with the adaption of the single-layer training scheme to multiple layers. More discussion is given in the supplementary document.

With the 2-layer LFNN, the MNIST classification task shows a 94.35% prediction accuracy, and its FNN counterpart shows 94.52% accuracy. As a reference, a 2-layer digital dense neural network, connected by the ReLU activation function achieves 98.32% accuracy.

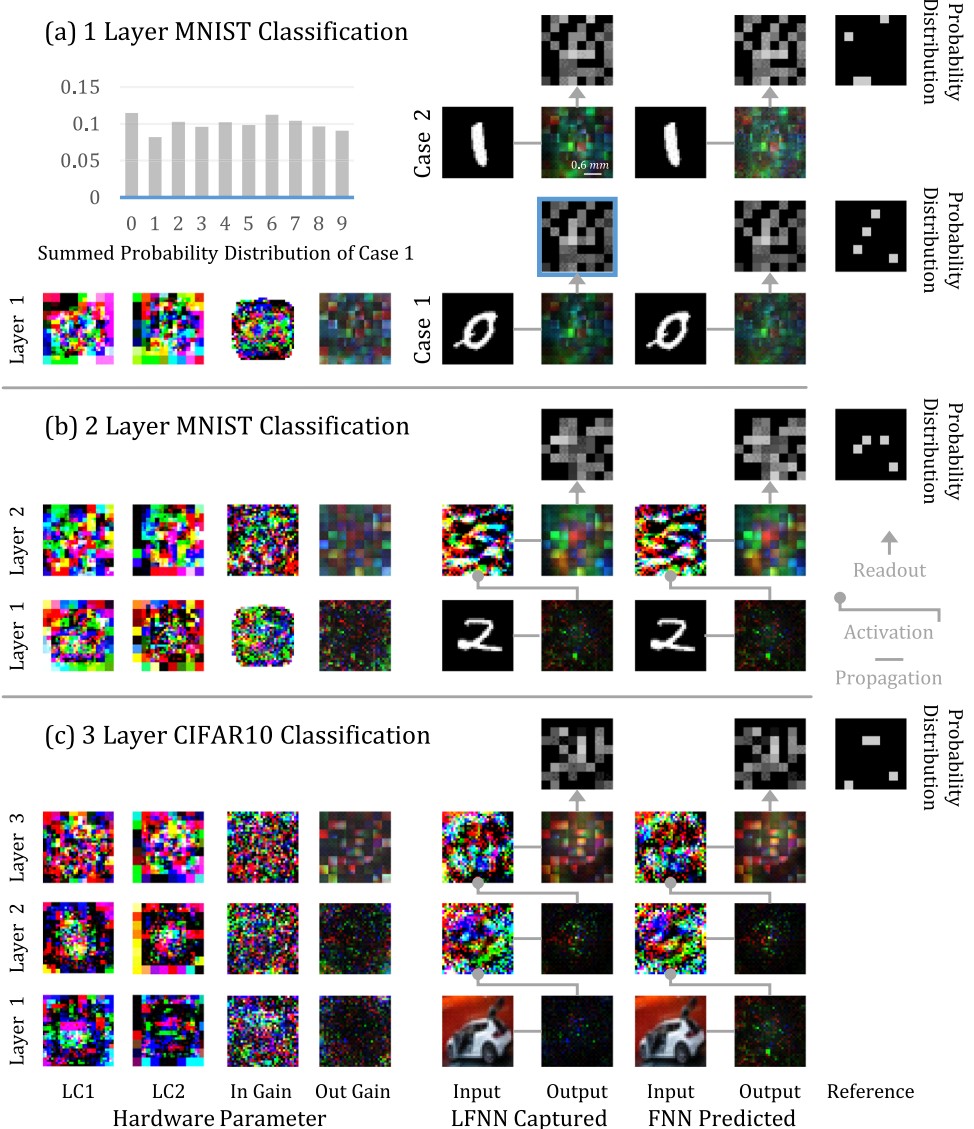

**Fig. 4 | Classification experiment.** We use the MNIST and CIFAR10 data sets to test one to three layers of the LFNN and visualize the results in (**a**), (**b**) and (**c**), respectively. The hardware parameters, FNN predicted outputs, and LFNN captured outputs are visualized together, layer by layer. For the classification task, the outputs of the last layer are chunked as probability distributions for different classes. We make the distribution layout trainable to adapt to the hardware optics automatically. For example, the areas corresponding to a class's probability are trained to disperse across the output plane to gather energy from the corners and then summed up as the final probability of that class. The readout is simply adding up pixels' RGB values as a probability.

Because the 2-layer LFNN achieves a high accuracy for the MNIST data set, we use a more challenging dataset containing color real-world objects, the standard CIFAR10[13], to evaluate a 3-layer LFNN. The FNN predicted accuracy is 46.19%, and the LFNN captured accuracy is 45.62%. As a reference, a 3-layer digital dense neural network achieves 53.52% accuracy. In the neuron array simulation, increasing the LC neuron size can significantly increase the accuracy of ONNs, nearly matching that of the digital dense neural network (Table 2a). Moreover, the state-of-the-art result in this dataset is achieved by deeper layer CNNs, and it is possible to apply similar ideas to ONNs by training multiple kernels in the future.

**Object recognition**
In addition to classification, we further evaluate the LFNN device with different tasks. One simple application is to use the device to recognize a specified object at the speed of light (Fig. 5). Here we use the MNIST and CIFAR10 datasets to train the device to recognize a specific class, e.g., the number zero or plane, among all classes, where the output plane is divided into positive and negative response areas for reading results. We test the 1-layer LFNN. For the recognition of the number zero with the MNIST dataset, the recognition accuracy reaches 98.12%. For the recognition of planes out of other objects, the accuracy is 77.30%.

**Depth estimation**
Finally, we apply a 4-layer LFNN to a depth estimation task using the 'coffee mug' category of the RGB-D object dataset[14]. Although the result suffers from certain detail loss due to the limited number of LC neurons, it demonstrates a potential application of an LFNN as a programmable light-speed sensor/lens (Fig. 6).

Besides application tests, we verify the feasibility of training random physical neurons by randomly disabling certain percentages of the LC neurons of the LFNN device to form Bernoulli arrays Table 2b. The result highlights the robustness of training loose neuron arrays with the FL paradigm, given the reasonable classification accuracy even with up to 60% of malfunctioning

## Table 1 | Comparisons against existing solutions

| (a) | Forward Model | Genetic Algorithm | Finite Difference | Functional Learning |
|---|---|---|---|---|
| **Gradient** | explicit | none | measured | implicit |
| **1-layer MNIST** | 23.50% | 14.06% | 8.594% | 90.78% |

| (b) | Data | Network | Mechanism | Applications |
|---|---|---|---|---|
| **Supervised Learning** | labeled | explicit | target learning | regression and prediction |
| **Unsupervised Learning** | unlabeled | explicit | self-structuring | data structure learning |
| **Reinforcement Learning** | decision process | explicit | trial-and-error | decision making |
| **Functional Learning** | z: functional response | z: explicit | z: induction | hardware, chip, |
| | p: labeled | p: implicit | p: deduction | and system control |

**(a)** Classification accuracy of training the 1-layer LFNN in the MNIST dataset using different training paradigms. The 1-layer LFNN consists of 2 LC panels. Genetic algorithm and finite difference are measured using approximately equal training time of functional learning. The forward model is measured using an equal number of epochs of functional learning. Detailed discussion is in the supplementary document.
**(b)** Different intuitions of existing learning paradigms.

## Table 2 | Experiments on various neural arrays

| (a) | Regular-2 | Regular-3 | Normal-3 | Uniform | LFNN | FNN | Digital DNN |
|---|---|---|---|---|---|---|---|
| **LC Neurons** | 2048 × 3 | 3072 × 3 | 3072 × 3 | 3072 × 3 | 2048 × 3 | | N.A. |
| **1-layer MNIST** | 91.03% | 92.07% | 92.40% | 92.45% | 91.02% | 91.39% | 92.71% |
| **2-layer MNIST** | 96.61% | 97.30% | 97.55% | 97.65% | 94.77% | 95.45% | 98.32% |
| **3-layer CIFAR10** | 47.48% | 50.61% | 51.73% | 52.53% | 45.62% | 46.19% | 53.62% |

| (b) | Bernoulli-0 | Bernoulli-20 | Bernoulli-40 | Bernoulli-60 |
|---|---|---|---|---|
| **LC Neurons** | 2048 × 3 | 1638 × 3 | 1229 × 3 | 819 × 3 |
| **Simulation** | 89.80% | 82.07% | 79.17% | 77.29% |
| **LFNN** | 89.13% | 81.83% | 79.36% | 75.65% |

**(a)** Classification accuracy of four neuron array simulations, the actual LFNN captured output, the FNN predicted LFNN output, and the equal-layer digital dense neural network (DNN) reference. The LFNN has 12, 288 trainable variables per layer, including 6, 144 to control LC neurons and 6, 144 to control the input/output intensity gains. The FNN has 28, 438, 144 trainable variables per layer. The digital dense neural network comprises dense layers connected by ReLU with neuron sizes of (784, 10), (784, 784, 10), and (3072, 3072, 3072, 10), respectively.
**(b)** Physical assessment of random neuron arrays. The test is conducted in the numerical simulation and the actual LFNN device for 1-layer MNIST classification. The Bernoulli arrays (Bernoulli-) are modified from the regular-2 array by randomly disabling 0% to 60% of LC neurons as well as all input/output intensity gains. The figure on the right visualizes a Bernoulli-60 array, where 60% of random LC neurons are disabled.

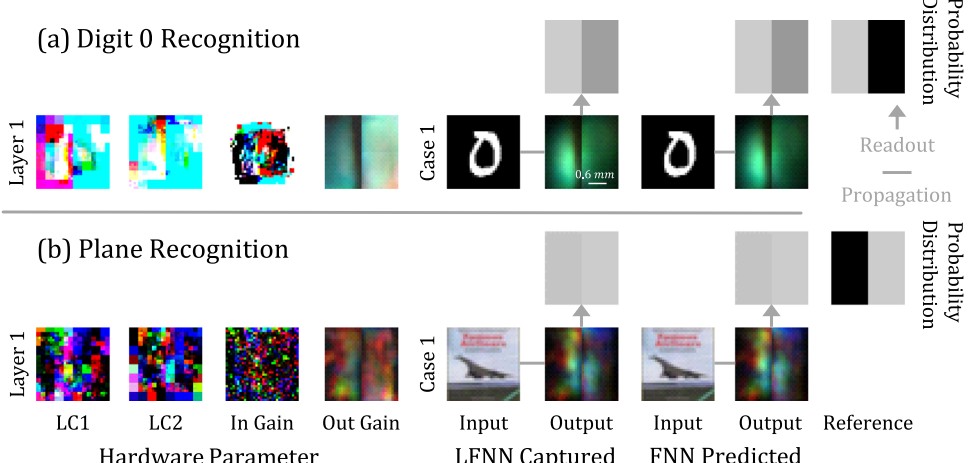

**Fig. 5 | Object recognition experiment.** We train the 1-layer LFNN to recognize digit 0 and planes within the MNIST and CIFAR10 data sets and visualize some of the results, including the hardware parameters, FNN predicted outputs, and LFNN captured outputs. **a** For digit 0 recognition, the output plane left and right halves are masked as positive and negative response areas. **b** The layout of the plane recognition task is the opposite. Readout is simply adding up pixels RGB values as a probability. Note that the trained probability distribution varies from task to task.

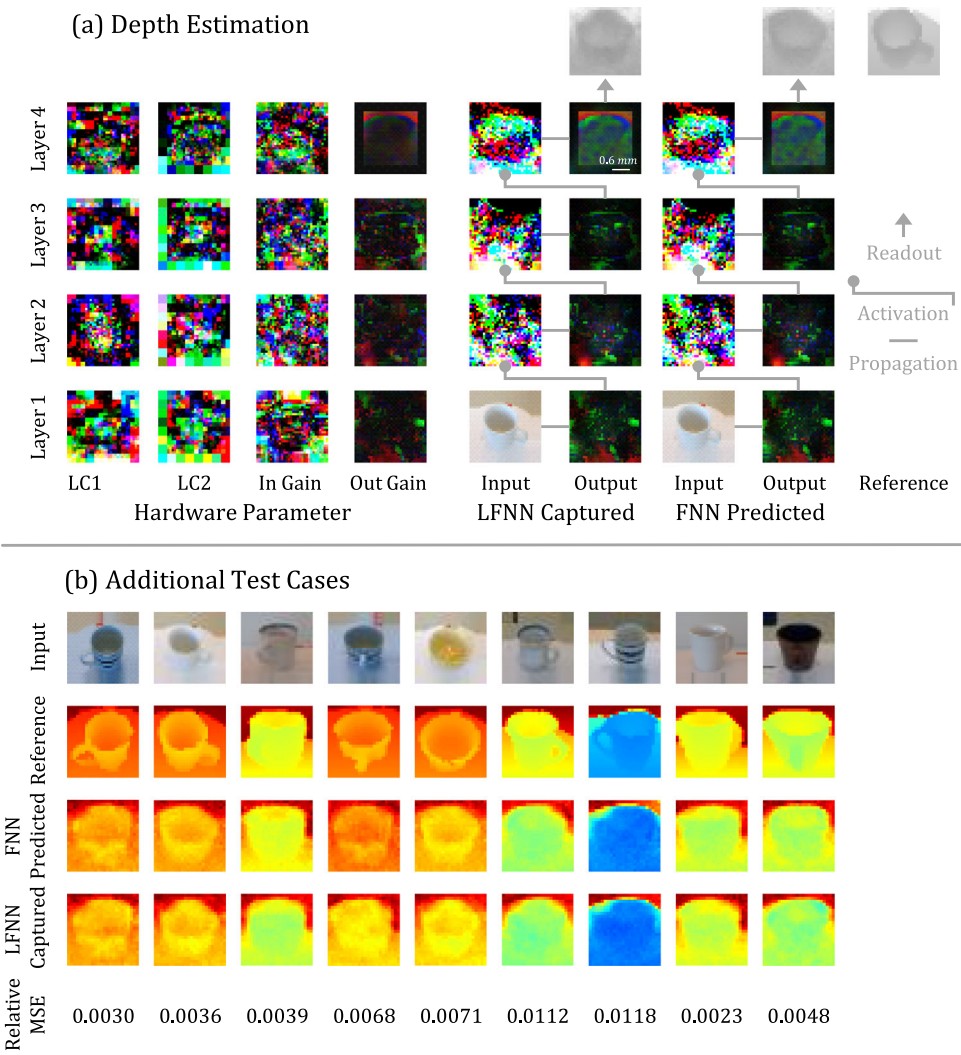

**Fig. 6 | Depth estimation experiment.** We train the 4-layer LFNN to predict the depth of RGB images of 'coffee mug'. The train set contains 4500 images, and the test set contains 300 images. **a** The RGB channels of the last layer are summed up as the depth output with a average relative mean squared error of 0.0073. **b** For the additional test cases, we use an Opencv color table called `COLORMAP_JET' to visualize the output depth. The readout is simply adding up pixels' RGB values as depth.

neurons. More experiment results and discussions are given in the supplementary document.

## Discussion

We find that even when using a naive hardware solution, the FL paradigm trains the uncalibrated LFNN well to enable programmable ONNs and to show results comparable to those of digital neural networks. While the hidden physical factors and noise continue to hinder the optimization, manufacturing, and applications of precision optics, this learning-based paradigm shows promising directions by demonstrating that high-fidelity inference results can be achieved by training loose neuron arrays consisting of non-differentiable, inexpensive, low-precision, and casually installed hardware solutions. In other words, this approach releases hardware fabrication from expensive high-fidelity components and exhausted precise assembling.

Another interesting observation is that the most general neuron array configuration, the uniform array, achieves the best performance in the numerical simulation (Fig. 1). Intuitively, the neuron array can be an extension of deep neural networks from software to hardware. It has been proven that stacking simple artificial neurons can outperform handcrafted algorithms in many domains. This observation confirms

our initial guess that non-handcrafted designs can replace handcrafted designs and even achieve unprecedented promising results. The non-handcrafted design not only releases us from many realization difficulties, but also pushes the limits of possible hardware designs, chip manufacturing methods and system control approaches, as such neuron arrays allow freedom in the design space.

ONN is promising for low-latency, high-bandwidth, and power-efficient neural network hardware. We realize a programmable incoherent ONN, LFNN, to verify the feasibility of using a group of loose neurons for practical neural network inference tasks. The LFNN can work as a general-purpose optical computing unit. In addition, it can directly process visible incoherent light, leading to possible applications for programmable lenses or computational displays, light-speed detectors for autonomous driving, and an optical-based X-Reality system.

The current LFNN prototype has achieved comparable results with equal-layer digital dense neural networks but with fewer trainable parameters by orders of magnitude, as shown in Table 2a. According to the numerical simulation (Fig. 1), the accuracy of ONNs nearly matches that of digital neural networks with more neurons. The current constraint of our practical implementation is the lack of computational resources to train more neurons.

Doubling the input and output resolutions leads to a 16-fold increase in the number of light field connections, which requires more computation resources or improved algorithms for training. Besides, the training of more ONN layers requires more time to gather *z-data* or reach a converged state. Because we use a commercial low-speed camera (4 fps) to implement the LFNN prototype, the average training time is 4 minutes per epoch. For example, it takes around 400 minutes to train the 100 epochs (Table 1a). Other training epochs are presented in Supplementary document. Although our method significantly outperforms existing approaches using equal training time, it could still be time-consuming. These difficulties can be resolved by developing better FL paradigms and updating to high-speed optical components. Finally, the state-of-the-art digital neural networks are advanced neural network structures like CNN and ResNets[15], but the ONN is still on its way to discovering its counterparts.

Further reducing noises can be crucial to improve the LFNN performance. In our hybrid design, noise mainly comes from both the optical and electronic components. On the one hand, strictly shielding the device to prevent environmental noise and introducing an additional cooling system to prevent thermal noise are the most straightforward approaches. Alternative possible optimizations include updating to stabler laser light sources or laboratory-standard LC panels. On the other hand, turning to a specific circle design with engineering techniques like grounding, shielding, and voltage stabilization are the common methods to reduce electron noises. Besides, shifting to professional sensors designed for low-light conditions might be a worthy attempt.

Many active research directions can help connect multiple layers of LFNN and perform all-optical nonlinear activation. For example, dye, semiconductor, or graphene saturable absorbers could be integrated with LFNN and X-activation in the future[16–23]. A promising solution is to combine memristance to build analog photoelectric activation hardware, which can handily realize various activation operators by nanosecond[24,25]. An economical option can make use of off-the-shelf high-speed cameras and micro LEDs[26–28].

The relationships and differences between the FL and some existing paradigms are summarized in Table 1a. Fundamentally, the FL paradigm implicitly trains an unknown function through an FNN to achieve the impossible. Both functional learning and reinforcement learning gather data while training, but the underlying learning mechanisms are different. While reinforcement learning follows the trial-and-error methodology, functional learning is essentially an induction-and-deduction process, where *z-learning* is the inductive reasoning of the hidden function with functional response data and *p-learning* is the deductive reasoning using the hidden function. As such, functional learning has higher efficiency in the utilization of data compared to reinforcement learning.

Currently, the FN paradigm is time-consuming for two reasons. First, *z-learning* gathers training data by capturing the actual outputs of the system. Therefore the training speed is majorly dependent on the system's response speed. Orders of speed-up can be expected by replacing the low-speed camera of the LFNN prototype with high-speed sensors for product-ready systems. Second, FNN has 28M parameters per layer and a much larger complexity than LFNN, implying considerable room to optimize the FNN network for efficient training. As a proof of concept, we train and test the LFNN tested in a closed environment using incoherent light sources. Realizing the direct inference of light signals of real-world objects is a more complicated challenge, which is worth a thorough follow-up study starting from dataset collection.

## Methods
### Functional learning
The optimization of the parameters of a model-free system is challenging. In order to train the hardware parameters of the LFNN, we propose a deep-learning paradigm, functional learning. Specifically, we generalize the learning problem as

$$\min_p \quad \|f(x;p) - y\|, \tag{3}$$

where $x$ is the input, $y$ is the target output, $p$ is the target trainable parameters, $\|\cdot\|$ denotes a general distance metric, and $f(\cdot;p) \in \mathcal{C}: [0,1]^{|x|} \to [0,1]^{|y|}$ is an unknown function in the function space mapping the continuous input $x$ to the target $y$. Because $f(\cdot;p)$ has no explicit formulation, we cannot mathematically solve the original problem of Equation (3). Therefore, we assume $f(\cdot;p)$ is an implicit function that can be represented by a functional neural network (FNN) as an instance with a learnable functional space. We then approximate the original problem with two sub-problems,

$$\min_{p,z} \|f(x;p) - g_N(z)(x;p)\| + \|g_N(z)(x;p) - y\|, \tag{4}$$

where $g_N(\cdot)(\cdot;\cdot)$ is a FNN (Functional Neural Network), $z$ is the functional space coefficients, and $p$ is the primary parameter, i.e., the hardware parameter in our case. However, both sub-problems are under-determined if we simultaneously optimize the functional space and the parameters that alter the mapping between input and output. Therefore we reformulate the equation as the following, by decoupling the variables to make each sub-problem solvable:

$$\min_{p,z,p_0,z_0} \quad \|f(x;p_0) - g_N(z)(x;p_0)\| + \|g_N(z_0)(x;p) - y\|$$
$$\text{s.t.} \qquad p_0 = p, z_0 = z, \tag{5}$$

where $p_0$ and $z_0$ are auxiliary variables. In this formulation, the problem can be solved by alternating minimization, which has shown convergence for non-convex problems[29,30], of each sub-problem. Specifically, at each epoch $k$, we update $p_0 \leftarrow p^k$ and $z_0 \leftarrow z^{k+1}$. The update rules can be then simplified as follows:

$$z^{k+1} \leftarrow \arg\min_z \|f(x;p^k) - g_N(z)(x;p^k)\|, \tag{6}$$

$$p^{k+1} \leftarrow \arg\min_p \|g_N(z^{k+1})(x;p) - y\|. \tag{7}$$

We propose different deep learning schemes to minimize sub-problems of Equation (6) and Equation (7), denoted as *z-learning* and *p-learning*, respectively.

**Algorithm 1. Functional Learning (FL) Paradigm.**
1: **function** FUNCTIONAL LEARNING
2:
3: Initialize $z$ and $p$ ▷ Initialize.
4: Initialize *p-optimizer* and *z-optimizer* using $g_N$
5: *p-data* ← $(x,y)$
6: $k \leftarrow 1$
7:
8: **while** $k <$ maximum epoch **do**
9: $u \leftarrow$ *p-data*. DRAWSAMPLE$(x)$
10: $v \leftarrow$ *LFNN*. CAPTURE$(u)$
11: *z-data* ← $(u,v)$ ▷ Update z-data.
12: *z-optimizer*. TRAINNETWORK$(z, z$-*data*$)$ ▷ Conduct z-learning.
13: *p-optimizer*. TRAINNETWORK$(p, p$-*data*$)$ ▷ Conduct p-learning.
14: $k \leftarrow k+1$
15: **end while**
16:
17: **end function**

**Learning Paradigm.** The FL paradigm departs from existing deep learning paradigms in several ways. The FNN has two sets of trainable

variables indicated here in blue and orange (Fig. 3a). The blue nodes (LC parameters) and blue lines (input and output plane intensity gains) are primary neurons that correspond to the trainable hardware parameters of the LFNN, termed *p-variable*. The orange lines and the variables of convolutional layers, termed *z-variable*, are trainable functional space coefficients trained to replace the hidden function $f(\cdot;p)$. The *p-* and *z-variables* are iteratively updated through *p-learning* and *z-learning*. *P-learning* accepts task-related data sets, e.g., CIFAR10, to optimize the primary parameter. *Z-learning* dynamically gathers *z-data* by sending impulses to the LFNN device to capture ground truth (GT) responses. The gradients are computed through backpropagation algorithms, and the variables are updated using the Adam optimizer[31] in the PyTorch framework.

Algorithm 1 shows the pseudocode of the FL paradigm. Functional learning (FL) uses a weaved learning scheme, which iteratively conducts *z-learning* and *p-learning*. Each learning procedure has independent loss functions, optimizers (*z-optimizer* and *p-optimizer*), trainable variables (*z-variablez* and *p-variablep*), and data set (*z-data* and *p-data*). First, we uniformly randomize the variables *z* and *p* in [0, 1]. Through the FNN $g_N$, *p-optimizer* calculates the gradients of *p* using backpropagation and updates the variables *p* using the Adam algorithm in the *p-learning* stage, and *z-optimizer* is the counterpart for *z*. At each epoch, *z-learning* and *p-learning* are conducted once. The *z-data* used for *z-learning* is updated at each epoch by randomly drawing certain input samples from *p-data* and capturing the corresponding responses of the LFNN device.

**P-learning**. *p-learning* trains the parameter neurons *p*, i.e., the hardware parameters for the LFNN device, to optimize the mapping from *x* to *y* in a specific function defined by the frozen functional space variables *z*. The training data set $p - data$ and the loss function depend on the chosen inference task. We use the Adam optimizer with a 0.001 learning rate.

**Z-learning**. *z-learning* trains the functional space variables *z* to approximate the unknown function *f* with $g_N(z)$ given a fixed *p*. Different from classic deep learning, the training data set *z-data* is updated at each epoch. We give some input signals to the unknown function *f*, i.e., the LFNN device in our case, and capture its responses. The input signals and output responses constitute the inputs and targets of *z-data*. Because we use an off-the-shelf low-speed camera to implement our LFNN prototype, capturing the responses of all *x* of *p-data* is time-consuming, we only randomly draw a subset with 1024 samples (line 9 of Algorithm 1) every epoch and repeatedly train such a subset ten times while shuffling. This acceleration scheme works well experimentally across the training process. We use the $L1$ loss and the Adam optimizer with a 0.001 learning rate for *z-learning*.

**Multi-Layer Learning Paradigm**. For an *n*-layer LFNN, Equation (4) can be extended by connecting multiple FNNs as:

$$
\begin{aligned}
\min_{p_i, z_i} \sum_{1 \le i \le n} & \left\| f(x_i; p_i) - g_N(z_i)(x_i; p_i) \right\| \\
& + \left\| g_N(z_n)(x_n; p_n) - y \right\|, \\
& x_1 = x, \\
& x_i = g_N(z_{i-1})(x_{i-1}; p_{i-1}), \quad for \quad i > 1,
\end{aligned}
\tag{8}
$$

where $p_i$ and $z_i$ are the FNN variables on the *i*-th layer. In general, we approximate the problem by minimizing every error of single layers by alternating minimization.

Algorithm 2 is the pseudo code. Compared to the 1-layer FL, we conduct the *z-learning* on one layer at a time during each epoch. Because the change of the function in single layer might lead to a new global optimization point, we treat the $p - learning$ as an end-to-end global optimization process and update all the entire set $p_i$. While training the first layer, we directly draw input samples from *x* to

capture $z\text{-}data_1$, then feed the output to the LFNN device again to capture the next layer's *z-data*.

To speed up the convergence of multi-layer training, we apply fine-tuning once the loss no longer drops, where only the last layer's parameters are updated and all other layers are frozen.

**Algorithm 2. Multi-layer Functional Learning (FL) Paradigm.**

```
1: function MULTI-LAYER FUNCTIONAL LEARNING
2:
3:    Initialize {z_i} and {p_i} for 1 ≤ i ≤ n ▷ Initialize.
4:    Initialize p-optimizer and {z-optimizer_i} using g_N for 1 ≤ i ≤ n
5:    p-data ← (x, y)
6:    k ← 1
7:
8:    while k < maximum epoch do
9:      for j ← 1 to n do
10:        if j = 1 then
11:           u_j ← p-data. DRAWSAMPLE(x)
12:        else
13:           u_j ← v_{j−1}
14:        end if
15:        v_j ← LFNN. CAPTURE(u_j)
16:        z-data_j ← (u_j, v_j) ▷ Update z-data.
17:        z-optimizer_j. TRAINNETWORK(z_j, z-data_j) ▷ Conduct z-learning.
18:        p-optimizer. TRAINNETWORK({p_i}, p-data) ▷ Conduct p-learning.
19:        k ← k + 1
20:      end for
21:    end while
22:
23: end function
```

### Functional neural network

Figure 7 is the decomposition of the FNN. We use one FNN to represent a LFNN layer. The network contains two components: a functional basis block and functional space coefficients. The design of the functional basis block is inspired by the physical properties of the actual hardware, which will be detailed later. The functional space coefficients extend the functional basis to a non-linear space. The lines or neurons are classified into three categories by color. Blue represents the primary variables, *p-variable*, orange represents the functional space variables, *z-variable*, and black represents fixed connections or neurons that gather inputs with specified operators. *z-variable* and *p-variable* are iteratively updated by *z-learning* and *p-learning*, respectively.

The functional basis block consists of various neurons inspired by the light field relationship between the input, LC, and output neurons. An input neuron can make a possible contribution to every output neuron and thus is connected to every output neuron to form a light field (Fig. 7b). The connections contain *p-variable* that represents the hardware parameter of the input plane's neuron-wise gain, and *z-variable* that is trained to optimize the functional space. An LC neuron may affect the connection between an input neuron and an output neuron. Thus, we connect every LC neuron to the connections of the light field between the input and output plane (Fig. 7c). The connections also contain *p-variable* that represents the neuron-wise hardware parameter for controlling the LC panel, and *z-variable* trained to optimize the functional space. The functional space coefficients block is five layers of a convolutional neural network (CNN) with a $3 \times 3 \times 64$ kernel and the rectifier linear unit (ReLU)[2]. The function is to non-linearly combine the functional basis block for fitting the non-linear part of the actual hardware. In addition, the output neurons are also weighted by the hardware parameter of the output plane's neuron-wise gain. The intuition is similar to that of an attention mechanism.

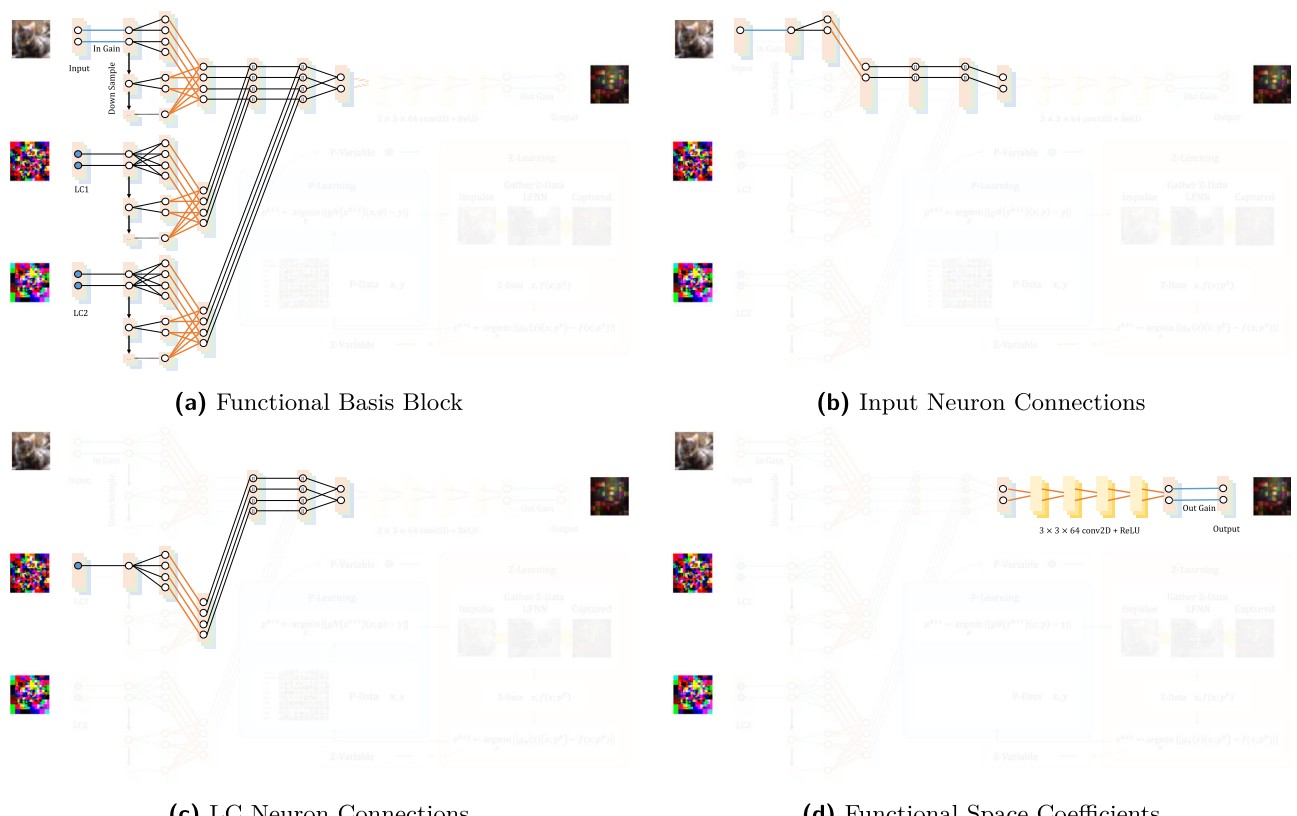

**(a)** Functional Basis Block          **(b)** Input Neuron Connections

**(c)** LC Neuron Connections          **(d)** Functional Space Coefficients

**Fig. 7 | Functional neural network (FNN) decomposition. a** The functional basis block reflects the connections of the neurons. **b** An input neuron has a connection to every output neuron. **c** An LC neuron potentially attenuates every connection between input and output neurons. **d** The functional space coefficients for the nonlinear combination of the function basis.

Because the signal passes through some relatively dim and noisy connections, we merge nearby neurons' connections to build stronger connections with less noise. As shown in Fig. 7a, we down-sample the input or LC plane half to combine nearby neurons and then merge the connections between different resolutions with the trainable variable, *z-variable*.

**Sparsification.** In practice, many connections of LC neurons, especially at the finest resolution, are very weak and therefore can be approximated by connections at coarser resolutions. Removing those connections from the FNN can save memory and accelerate the training. We measure the impacts of connections by alternating the LC neuron parameter between 0 and 1 and capturing the energy change on the output plane. If the energy change of a connection between an input and output neuron pair w.r.t. an LC neuron is below a threshold, we remove that LC neuron's connection to that pair. For the finest resolution, we experimentally set the threshold to 20% of the maximum energy of valid input and output neuron connections. For the second finest resolution, we experimentally set the threshold to 10%. For the coarsest resolution, we keep all connections.

Because there are billions of connections to test and conducting a one-by-one test is time-consuming, we build quad-trees in the neurons of an LC plane and hierarchically perform the test. Figure 3b shows a simple example of the quad-tree searching algorithm. If changing the parameters of LC neurons in a node yields an energy change larger than the threshold, we mark its sub-nodes for testing. Otherwise, we do not need to test any of its sub-nodes.

In summary, this sparsification scheme prunes 99.96% of the LC neuron connections for our LFNN implementation.

**Multi-Layer Connection.** In order to capture multiple-layer LFNN results, we cycle the captured output to the input of the next LFNN layer and conduct certain activation functions with the signal. These activation functions are currently performed on a computer but can be implemented by all-optical or non-digital photoelectric hardware in the future. First, we design a special activation, termed X-activation, to enhance the inference capability of the multi-layer ONN. Specifically, the captured output is subtracted by its 180-degree rotated image. The subsequent X-activation is a common neuron-wise batch normalization method[12]. Finally, a simple amplitude filter integrated into the driver of the input plane clamps the impulse between zero and one and also introduces non-linearity.

This design of X-activation aims for four different targets. First, it enables the deactivation in optical neural networks. The data flow of common artificial neuron networks contains both positive and negative values, but there is no negative light in an incoherent optical system. By introducing X-activation, a neuron can either activate or deactivate an arbitrary subsequent neuron with different LC neuron parameters (Fig. 3c). Second, the outputs of X-activation are positive-negative neuron pairs. Thus, one neuron in a pair would be nullified by the intensity filter and activation yields sparse input for the next layer with a stable activation ratio. Third, the mean of the output of X-activation is consistently 0, which cancels the shift in the data flow due to the attenuation effect of the optical components. Fourth, X-activation has only simple logic and operators that can be implemented by non-digital hardware in the future to match the activation to speed-of-light processing.

## Light field neural network

We physically prototype a loose neuron array, termed light field neural network (LFNN), to realize the regular-2 array and Bernoulli array by

randomly disabling neurons. We use off-the-shelf components, e.g., liquid crystal display (LCD) panels, polarizers, and a machine vision camera, without tedious calibration. As a result, the actual LFNN is not exactly the same as the regular-2 array.

The applied LCD panels are expected to show a high transmittance and good linearity between the applied voltage and the polarization rotation angle. Among available LCD panels, we use Chimei Innolux AT070TN83 as the optical layer. The photograph of our prototype is shown in Fig. 2. A modified backlight system adapted from a commercial projector is used to illuminate the input plane. The front and rear polarizing films are removed from the front of the two LCDs. A diffuser and a polarizing film are located at the camera's focal plane as the output plane. All layers are assembled into acrylic frames separately, and all frames are installed on the optical table. The distance setting between the layers is 30 mm. The output plane images were acquired by a machine vision camera (The Imaging Source DFK 33G274) with the pixel pitch of 4.40 $\mu m$, the resolution of $1600 \times 1200$, and the bit depth 12 bit. The focal length and F-number of the camera are 12.5 mm and 1.6, respectively. Because the device is not fine-tuned and the planes have different valid areas, we use the pixels in the center areas of these planes as neurons. Specifically, $15 \times 13$ pixels in the input plane, liquid crystal 1 (LC1), and liquid crystal 2 (LC2) are jointly controlled as one input neuron and LC neurons. $12 \times 15$ pixels in the camera's image plane are read together as one output neuron.

## Dataset

In our experiment, we use two standard classification datasets. The MNIST dataset contains 60,000 training images and 10,000 test images belonging to 10 classes of handwritten digits. From 0 to 9, each class contains 5,923(980), 6,742(1,135), 5,958(1,032), 6,131(1,010), 5,842(982), 5,421(892), 5,918(958), 6,265(1,028), 5,851(974), and 5,949(1,009) training(test) samples, respectively. The CIFAR10 dataset consists of 60,000 $32 \times 32$ color images in 10 classes of real-world objects, with 6,000 images per class. There are 5,000 training images and 1,000 test images in each class. The classes include airplane, automobile, bird, cat, deer, dog, frog, horse, ship, and truck.

## Data availability

The MNIST[8] data used in this study are available in the MNIST dataset [http://yann.lecun.com/exdb/mnist/]. The CIFAR10[13] data used in this study are available in the CIFAR-10 dataset [https://www.cs.toronto.edu/~kriz/cifar.html]. The RGB-D images[14] used in this study are available in the RGB-D dataset [http://www.cs.washington.edu/rgbd-dataset]. The processed training data are available in the repository of the project [https://github.com/huoyuchisc/FL]. Source data are provided with this paper.

## Code availability

The source codes of this study are available in the repository of the project [https://github.com/huoyuchisc/FL].

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

## Acknowledgements

We would like to thank all reviewers for their insightful comments. We also thank Liang XU for building the LFNN prototype. The project is supported in part by Key R&D Program of Zhejiang Province (No. 2023C01039, R.W), NSFC (No. 61872319, R.W.), the Fundamental Research Funds for the Central Universities (R.W.), PI funding of Zhejiang Lab (121005-PI2101, Y.H.), Key Research Project of Zhejiang Lab (No. K2022PG1BB01, Y.H.), the Hong Kong UGC Early Career Scheme Fund (27212822, Y.P.), and the Start-up Fund of the University of Hong Kong (Y.P.). This project is also supported in part by MSIT/NRF (No. RS-2023-00208506, S.Y.), ITRC (IITP-2023-2020-0-01460, S.Y.) of Korea.

## Author contributions

Y.H. conceived the idea and developed the theory. Yuchi Huo and Yifan Peng verified the analytical methods. Y.H. wrote the code. Y.H., Y.P., and R.W. conceived and planned the experiments. Y.H. and C.G. carried out the experiments. Y.H., S.-E.Y., and Y.P. wrote the manuscript. H.B. and S.-E.Y. supervised the project. All authors discussed the results and contributed to the final manuscript.

## Competing interests

The authors declare no competing interests.
