## [Peer review file · Nature Communications]

REVIEWER COMMENTS

Reviewer #1 (Remarks to the Author):

In this article named “Optical Neural Network via Loose Neuron Array and Functional Learning”, the authors report a functional learning method to train loosely connected neural networks, which was demonstrated using an optical neural network based on modulation of light field: light field neural network (LFNN). The proposed functional learning paradigm might be valuable for the training of machine learning inference hardware that operates based on a non-differentiable inference process or has unknown/uncalibrated physical properties or parameters.

Overall this is a solid manuscript with important results that would be of broad interest for Nat Comm readership. There are some questions and discussion topics that can further improve the manuscript:

1. The LFNN trained based on a perfect forward model can be regarded as the baseline against the new learning paradigm proposed by the authors. According to the paper, for this baseline model they achieved only 19.72% prediction accuracy in the experiment, which is significantly lower than the other results. This severe degradation of performance should be caused by the real-world “imperfections” such as noise, narrow view angles, etc. as the authors also mentioned. Without any knowledge of these “imperfections” the models are probably trained to be sensitive to them in the inference phase. But without further calibration, actually there is a feasible route to mitigate the effect of some of these “imperfections” by including them in the analytical forward models, e.g., noise, misalignment, and narrow viewing angles – for a similar strategy see for example DOI: 10.1515/nanoph-2020-0291. Some of these imperfections can be easily estimated by performing characterization experiments like what the authors did in the supplementary material, and the training with the estimation of these “imperfections” would probably save some major performance loss. The authors can further expand on this to discuss different possible options and their pros/cons. Besides, the configuration used for this 19.72% model is not clear - is it a regular-2 configuration or something else? And it would be better if more details of the analytical forward model could be added.

2. In this paper the authors used 3 tasks, i.e., 10-class image classification, image recognition and depth estimation for application tests. For the third task, depth estimation, the authors only show examples of results (and loss curves in supplementary materials). It would be better to evaluate the model using metrics such as RMS error. Also, some of the depth prediction results and ground truth seem to be dark, which makes it hard to judge the quality and correctness of the generated dataset or the trained model. The quantification of the success of this third task needs more clarity and discussion.

3. The authors used two LC panels with a spacing of 30 mm in their physical implementation. It would be good to explain how this distance was selected? Considering the LFNN has narrow scattering angles, why was a larger spacing not used to create a better connectivity between the adjacent LC panels? Besides, the authors included three LC panels in their numerical simulations, and the results are improved compared to using two. It would be better to provide a more complete analysis regarding effect of the number of LC panels and their spacing, so that the selection of these parameters for implementation can be justified.

4. The authors trained an FNN to approximate the behavior of their LFNN. According to the paper, LFNN has 12k parameters per layer, while the FNN has 28M parameters per layer and much larger complexity. Because of this large discrepancy in the number of parameters, there should be room to further prune/simplify the structure of this large FNN for more efficient training. Once achieved, it would be easier and faster to train a LFNN with more neurons. More analysis/discussion here could be beneficial.

5. The authors performed various optical characterization experiments/analyses of their LFNN, which is highly appreciated. It would be interesting if the authors show some similar but virtual characterization of their FNN, where probably the results will very much resemble LFNN but with slight differences. This can be a good supporting evidence to demonstrate that the FNN really learned the intrinsic properties of LFNN.

Other comments:

- In the uniform array configuration of LFNN neurons, what is the specific magnitude of the normal distribution disturbance? And what is the rationale for selecting this magnitude.
- The order of figures/tables seems to be out of order. For example, in the main text Figure 5 is mentioned immediately after Figure 2.
- In the main text, the authors mentioned there is a regular-2 array in Figure 1. But readers will not understand what “regular-2” is until they read the supplementary material. Also, it seems that the explanation of c, d, e in Figure 1 is missing the main text. Therefore, it would be better to remove c, d and e from Figure 1 but keep them in Figure S1. Also, the same comment applies to Table 1.
- Scale bars for the sensor measurement results in many figures are missing.
- There are multiple typos in the text. For example, In Line 244, Page 4, main text, “losse” should be “loose”. In Line 132, Page 3, main text, “pannel” should be “panel”.
- In Line 554 and 561, Page 8, main text, “Figure 6 (c)” and “Figure 6 (d)” should be “Figure 6 (b)” and “Figure 6 (c)”, respectively. In Line 666, Page 10, main text, “mum” should be “ μm ”.

Reviewer #2 (Remarks to the Author):

The authors propose an optical neural network via Loose Neuron Array and give a method of functional learning. The novel feature of this structure is that it utilizes simple, low-precision, loose neurons to realize a programmable optical neural network that directly processes visible incoherent light and can process RGB images. In addition, a functional learning method is designed to train the structure.

While the research appears to be detailed and accurate, the paper requires improvement and more works before it can be further considered for publication. My detail comments are as follows:

1. The proposed optical neural network is incoherent, but it doesn't reflect the usage and advantages of incoherent light in the content of the paper.
2. In the experiment, how to read out the outputs obtained by FNN prediction and LFNN capture?
3. In the third page, fourth page and Table 1 of main paper, the digital DNN can achieve 92.71%, 98.32%, and 53.62% accuracy in the classification for one-layer MNIST, two-layer MNIST and three-layer CIFAR10, respectively. Please provide references, experimental procedures and other evidences that can prove the results here.
4. In the classification experiments of MNIST with one- and two-layer neural networks and CIFAR10 with three-layer neural network, the probability distribution layout of each class is similar, only the depth of each small grid is different, so how is the discriminating mechanism of each class established? I hope you can open the source code to verify your experimental process.
5. In the classification task, the predicted category probability is obtained by training the probability distribution layout. Among them, it is proposed that because the input neurons of the LFNN prototype have a narrow scattering angles, some important features on the input plane cannot propagate the information to the output plane corner, so four dispersed connections are left to each class, but I think the reason not sufficient, you need to point out the more convincing reasons for this. Secondly, how to realize these four dispersed connections, how to determine the four dispersed connections corresponding to each category, that is, what is the criteria? In addition, I also noticed that for the same target classification of one- and two-layer neural networks, their four dispersed connections are not the same. Please explain the reason; if not, please make corrections.
6. The optical neural network model proposed by the author has a simple structure and low complexity. However, the inference performance of the model is closely related to the complexity of the model structure. For the one-layer and two-layer MNIST classification, 91.02% and 94.77% of high classification accuracy have been achieved; for the classification of CIFAR10, I observed that the classification accuracy of LFNN and FNN were only 45.62% and 46.19%, respectively. It can be seen that this system had poor effect in processing complex data sets and completing complex tasks.

Here, it is suggested that you can perform inference with more-layer neural networks such as four, five, and six layers for MNIST and CIFAR10, and observe whether the classification performance can be

effectively improved, especially the classification of CIFAR10, and provide demonstrable experimental results.

7. In the section 2.2 of Methods, X-activation designed in a multilayer LFNN is described. This activation is aimed at four different targets A, B, C and D, then how to judge the strength of the connection between x and A and D, and what is the significance of the existence of B and C neurons. I suggest to give an example to more clearly illustrate the working principle of X activation.

Moreover, X-activation is an electronic activation function, and it has only 5% effect on the three-layer CIFAR10 classification, and the classification performance is not improved significantly. We would like to see the nonlinear activation of optics, which is a leapfrog innovation for the realization of optical neural networks. And you also mentioned that X-activation can be implemented by non-digital hardware in the future to match the activation to speed-of-light processing. It is recommended that you propose a possible implementation method or technical approach, and implement and apply it in ONN as much as possible.

8. I suggest providing a supplemental document such as a multimedia file to show your hardware architecture, experiment processes and results.

Reviewer #3 (Remarks to the Author):

This paper proposes a function learning method to train the optical neural network without strict physical modeling of the system. An electronic convolution neural network model is employed for modeling the optical neural network. During the training, parameters of the electronic and optical neural networks are updated iteratively for the convergence of the system modeling and task learning. The idea seems fancy, however, I have strong concerns about the efficacy, efficiency, and scalability of the proposed method. Detailed concerns are:

1. (efficacy) Details of ablation analysis. The only comparison counterpart is ref. 9 of the main text. As stated in the main (line 167) and supplement (line 133), the proposed method can improve the testing accuracy on the MNIST test set from 19.72% to 90%+. However, few details are provided about the implementation of the comparison. If there is a revision process, I suggest the authors add details about the comparison process, especially the “point spreading functions”, the “analytical model” (line 130 supplement), and the input and output examples along with expectations in the supplement.

2. (efficacy) Other comparison recommendations. There are many training methods without accurate interrogation of physical parameters for optical neural networks, which I will recommend for proof of the efficacy of the proposed method. Some examples are gradient calculation through finite-difference (ref. 6 main texts) and genetic algorithm (<https://doi.org/10.1021/acsphotonics.1c00035>).

3. (efficiency) In the supplement, the training of a neural network with the proposed method takes as many as thousands of epochs.

Since each epoch takes about 4 minutes, it would take several days to train the neural networks in this work, which is quite long considering the scale of the network. I suggest the authors add some discussions on the efficiency of the proposed training method.

4. (scalability) In this work, the network scale are limited to four-layer amplitude-only modulation, and the capability is limited to binary classification, both of which are not close to real-world tasks. I suggest the authors add numerical exploration on more advanced tasks like 10-class classification with a larger number of layers, which is actually still quite easy tasks compared with an electronic neural network.

5. (scalability) . As an optical computing method in free space, I think they have a similar major problem to other optical neuromorphic computing methods. How do the authors plan to scale their system (which uses conventional and mature optical components) to a form factor and performance which is relevant for the field of AI and machine learning? The computing density is still very limited due to the liquid crystal used in this method. The current computing performance, as described in my previous questions, seems very limited compared with other optical computing methods or electronic methods.

6. (application) Incoherent optical computing is a very exciting and challenging area, as it's very hard to model the light propagation with a large density. As the authors claim incoherent optical neural network, I would expect they show practical applications on normal scenes, e.g. object recognition in real-world, or depth estimation for a 3D sample. I think such an experiment will significantly improve the impact of this work.

We would like to thank all reviewers for the detailed comments and suggestions. In the revision, we have conducted a lot of new experiments, such as the comparisons with finite difference and genetic algorithm as proposed by Reviewer #1 and Reviewer #3. The raised comments are carefully addressed. While there are some raised extensions of researches that cannot be done in this paper, we discuss our future research plans.

We first summarize our major modifications:

1. We conduct comparisons between existing training paradigms, including explicit forward model, finite difference, genetic algorithm, and our functional learning. The results are reported in Table and discussed in the manuscript and supplementary document.
2. We conduct experiments in regards to the effect of neuron number and layer spacing with numerical simulation. The results are reported in the supplementary document (Table S2 and Section S1.1).
3. We verify that the functional neural network can reflect the actual optics of the light field neural network with visualized differences in Section S3.4.
4. We use pseudo-color to enhance the visualization of network output in Fig.9 and report the numerical errors.
5. We improve the exposition with additional contents, such as explanations of 'Readout' in Figure 7-9, a probability histogram in Figure 7, and additional discussion of the X-activation in Figure 4.
6. We add discussions of limitations and future plans to the end of the manuscript.
7. We attach a video to show the device.

In the following, we present our point-to-point responses to the concerns.

Reviewer #1 (Remarks to the Author):

In this article named "Optical Neural Network via Loose Neuron Array and Functional Learning", the authors report a functional learning method to train loosely connected neural networks, which was demonstrated using an optical neural network based on modulation of light field: light field neural network (LFNN). The proposed functional learning paradigm might be valuable for the training of machine learning inference hardware that operates based on a non-differentiable inference process or has unknown/uncalibrated physical properties or parameters.

Overall this is a solid manuscript with important results that would be of broad interest for Nat Comm readership. There are some questions and discussion topics that can further improve the manuscript:

1. The LFNN trained based on a perfect forward model can be regarded as the baseline against the new learning paradigm proposed by the authors. According to the paper, for this baseline model they achieved only 19.72% prediction accuracy in the experiment, which is significantly lower than the other results. This severe degradation of performance

should be caused by the real-world “imperfections” such as noise, narrow view angles, etc. as the authors also mentioned. Without any knowledge of these “imperfections” the models are probably trained to be sensitive to them in the inference phase. But without further calibration, actually there is a feasible route to mitigate the effect of some of these “imperfections” by including them in the analytical forward models, e.g., noise, misalignment, and narrow viewing angles – for a similar strategy see for example DOI: 10.1515/nanoph-2020-0291. Some of these imperfections can be easily estimated by performing characterization

experiments like what the authors did in the supplementary material, and the training with the estimation of these “imperfections” would probably save some major performance loss. The authors can further expand on this to discuss different possible options and their pros/cons. Besides, the configuration used for this 19.72% model is not clear - is it a regular-2 configuration or something else? And it would be better if more details of the analytical forward model could be added.

- The configuration is regular-2. We update the discussion in the main manuscript and add additional details about the analytical forward model in the supplementary document (Sec S3.3). In addition, we also evaluate two other training paradigms, finite-difference and genetic algorithm as suggested by Reviewer #3.

2. In this paper the authors used 3 tasks, i.e., 10-class image classification, image recognition and depth estimation for application tests. For the third task, depth estimation, the authors only show examples of results (and loss curves in supplementary materials). It would be better to evaluate the model using metrics such as RMS error. Also, some of the depth prediction results and ground truth seem to be dark, which makes it hard to judge the quality and correctness of the generated dataset or the trained model. The quantification of the success of this third task needs more clarity and discussion.

- We report the numerical errors and use pseudo color to enhance the visualization of output depth in Figure 9.

3. The authors used two LC panels with a spacing of 30 mm in their physical implementation. It would be good to explain how this distance was selected? Considering the LFNN has narrow scattering angles, why was a larger spacing not used to create a better connectivity between the adjacent LC panels? Besides, the authors included three LC panels in their numerical simulations, and the results are improved compared to using two. It would be better to provide a more complete analysis regarding effect of the number of LC panels and their spacing, so that the selection of these parameters for implementation can be justified.

- To determine the distance, we first set the distance between the input plane and the output plane as 90~mm so that the central neuron's energy distribution can roughly cover the entire output plane. Then we equally split the spacing to insert two LC layers. Besides, we add additional experiments in Sec S1.1 and Table S2 to evaluate the

impact of spacing and neuron numbers.

4. The authors trained an FNN to approximate the behavior of their LFNN. According to the paper, LFNN has 12k parameters per layer, while the FNN has 28M parameters per layer and much larger complexity. Because of this large discrepancy in the number of parameters, there should be room to further prune/simplify the structure of this large FNN for more efficient training. Once achieved, it would be easier and faster to train a LFNN with more neurons. More analysis/discussion here could be beneficial.

- We add discussion about how to improve training efficiency at the end of the main manuscript.

5. The authors performed various optical characterization experiments/analyses of their LFNN, which is highly appreciated. It would be interesting if the authors show some similar but virtual characterization of their FNN, where probably the results will very much resemble LFNN but with slight differences. This can be a good supporting evidence to demonstrate that the FNN really learned the intrinsic properties of LFNN.

- We add additional results in Sec. S3.4 to evaluate the accuracy of the FNN outputs.

Other comments:

- In the uniform array configuration of LFNN neurons, what is the specific magnitude of the normal distribution disturbance? And what is the rationale for selecting this magnitude.

- We add discussions about the magnitude of the distribution disturbance in Sec S1.1. Specifically, for the normal-3 array, we first draw 3 random numbers from a normal distribution of standard deviation 0.25 to distribute each neuron within a unit cube with 99.99% confidence. Then the cube is resized to match the actual physical size of the whole system, where the two axes on the LC plane are scaled to pixel width and the last axis is scaled to one-third of the spacing between two LC layers. In such a case, the overlap between the neurons of the same LC plane is maximized. For the offset between two LC layers, we make the offset relatively larger to simulate misalignments. For the uniform array, all 3072 x 3 LC neurons are uniformly distributed between the first and the last LC layers. For each neuron, we draw a random number from a uniform distribution to determine its distance to the first and the last LC layer while keeping the other two axes regularly aligned as the regular-3 array.

- The order of figures/tables seems to be out of order. For example, in the main text Figure 5 is mentioned immediately after Figure 2.

- The reason for the misleading order is that Figure 5 belongs to the main manuscript and is too large to be placed in the middle of the manuscript. Therefore, it was supposed to be at the end of the paper. However, Figures 3 and 4 belong to the

Appendix Method and should be placed before the end of the paper. We will check the official format with editors later to arrange them properly.

- In the main text, the authors mentioned there is a regular-2 array in Figure 1. But readers will not understand what “regular-2” is until they read the supplementary material. Also, it seems that the explanation of c, d, e in Figure 1 is missing the main text. Therefore, it would be better to remove c, d and e from Figure 1 but keep them in Figure S1. Also, the same comment applies to Table 1.
- We add an introduction of the four neuron arrays in the caption of Figure 1. If that is not enough, we are glad to put the content about neuron arrays in the supplementary document.
- Scale bars for the sensor measurement results in many figures are missing.
- We add scale bars to Figure 7, 8 and 9. The scale bars for all other results are the same.

Reviewer #2 (Remarks to the Author):

The authors propose an optical neural network via Loose Neuron Array and give a method of functional learning. The novel feature of this structure is that it utilizes simple, low-precision, loose neurons to realize a programmable optical neural network that directly processes visible incoherent light and can process RGB images. In addition, a functional learning method is designed to train the structure.

While the research appears to be detailed and accurate, the paper requires improvement and more works before it can be further considered for publication. My detail comments are as follows:

1. The proposed optical neural network is incoherent, but it doesn't reflect the usage and advantages of incoherent light in the content of the paper.

- As a proof of concept, we train and test the LFNN in a closed environment using incoherent light sources to verify the possibility of the direct inference of real-world objects' signals. However, it is systematic work that requires a lot of additional efforts. Your suggestion is one of our next steps and we specifically describe it in a new discussion at the end of the paper
-

2. In the experiment, how to read out the outputs obtained by FNN prediction and LFNN capture?

- Since we use a commercial camera to capture the LFNN's output and run FNN on a PC, we directly parse the digital output from a computer. For example, the classification task parses the image to probability distribution by linearly adding up pixels' RGB value; and similarly, depth is also parsed by linearly adding up pixels' RGB value. We clarify

it in the revised version.

3. In the third page, fourth page and Table 1 of main paper, the digital DNN can achieve 92.71%, 98.32%, and 53.62% accuracy in the classification for one-layer MNIST, two-layer MNIST and three-layer CIFAR10, respectively. Please provide references, experimental procedures and other evidences that can prove the results here.

- As have been clarified in the caption of Table 1, the digital dense neural network comprises dense layers connected by batch normalization and ReLU with neuron sizes of $(784,10)$, $(784,784,10)$, and $(3072,3072,3072,10)$, respectively. That choice of neuron size depends on the input images. Specifically, the MNIST input is monochrome images of 28x28 neurons, and the CIFAR10 input is RGB image of 32x32x3 neurons. Note that we keep the hidden neuron size the same as the input neuron size, which is also the case of the LFNN device. The experiments are conducted based on the official PyTorch tutorials code without CONV2d layer to make a fair comparison with the fully-connected training scheme of the FNN.
- The tutorials can be accessed from the PyTorch official site:
https://pytorch.org/tutorials/beginner/blitz/cifar10_tutorial.html
- Here we also attach our source code in the appendix.

4. In the classification experiments of MNIST with one- and two-layer neural networks and CIFAR10 with three-layer neural network, the probability distribution layout of each class is similar, only the depth of each small grid is different, so how is the discriminating mechanism of each class established? I hope you can open the source code to verify your experimental process.

- Each category sums up the RGB values of four correlated spots as the predicted probability. The corresponding spots for each category are illustrated in Sec. S4.1 and S4.2. Even though the output patterns have similar for different inputs, summing up the outputs' RGB values yields discriminative probability distribution that can distinguish the probabilities of various categories. The process is only simple adding. We add a figure to illustrate the summed probability of Case 1 in Figure 7. As can be seen, summing the values of the four spots corresponding to digit 0 yields a probability of 11.47%. Similarly, summing the output of Case 1 on the four spots for digit 1 gets a probability of 8.208%.

5. In the classification task, the predicted category probability is obtained by training the probability distribution layout. Among them, it is proposed that because the input neurons of the LFNN prototype have a narrow scattering angles, some important features on the input plane cannot propagate the information to the output plane corner, so four dispersed connections are left to each class, but I think the reason not sufficient, you need to point out the more convincing reasons for this. Secondly, how to realize these four dispersed connections, how to determine the four dispersed connections corresponding to each category, that is, what is the criteria? In addition, I also noticed that for the same target

classification of one- and two-layer neural networks, their four dispersed connections are not the same. Please explain the reason; if not, please make corrections.

- Because manually optimizing the probability distribution layout requires an explicit model of the LFNN, there is no trivial algorithm to do so. We decide to make the probability layout distribution also trainable as suggested in the paper.
- As has been discussed in the Sec. S4 and illustrated in Figure S18, the connection is determined by a pruning process. The criteria to determine the connection is the trainable connection weights linking a point to a category. Because the whole process is totally trainable, the probability distribution layout is task-specific and trained by different datasets and LFNN architectures. The one- and two-layer neural networks represent two different hardware architectures and are naturally supposed to have different layouts. For example, the first LFNN layer extract features and has totally different scattering properties compared with the original input image, leading to different output layouts.
- We add more discussion in Sec. S4.

6. The optical neural network model proposed by the author has a simple structure and low complexity. However, the inference performance of the model is closely related to the complexity of the model structure. For the one-layer and two-layer MNIST classification, 91.02% and 94.77% of high classification accuracy have been achieved; for the classification of CIFAR10, I observed that the classification accuracy of LFNN and FNN were only 45.62% and 46.19%, respectively. It can be seen that this system had poor effect in processing complex data sets and completing complex tasks.

Here, it is suggested that you can perform inference with more-layer neural networks such as four, five, and six layers for MNIST and CIFAR10, and observe whether the classification performance can be effectively improved, especially the classification of CIFAR10, and provide demonstrable experimental results.

- The number of layers has strong correlation with the capability of a neural network. For difficult classification of CIFAR10, three layers produce relatively poor results for both digital and optical DNNs. We are highly interested in training more layers of LFNN, e.g., 6 to 10 layers, but the training speed is the primary difficulty that prevent us doing so. As discussed in the end of the main manuscripts, we will first improve the training speed to make dense-layer experiments more efficient.

7. In the section 2.2 of Methods, X-activation designed in a multilayer LFNN is described. This activation is aimed at four different targets A, B, C and D, then how to judge the strength of the connection between x and A and D, and what is the significance of the existence of B and C neurons. I suggest to give an example to more clearly illustrate the working principle of X activation.

Moreover, X-activation is an electronic activation function, and it has only 5% effect on the three-layer CIFAR10 classification, and the classification performance is not improved significantly. We would like to see the nonlinear activation of optics, which is a leapfrog

innovation for the realization of optical neural networks. And you also mentioned that X-activation can be implemented by non-digital hardware in the future to match the activation to speed-of-light processing. It is recommended that you propose a possible implementation method or technical approach, and implement and apply it in ONN as much as possible.

- The strength between an input neuron, says neuron x, and the neurons of the output plane are trained by FN and modulated by the LC panels as discussed in Method Sec 1 and 2. The X-activation only impacts the connections between the neurons on the output plane and the neurons on the input plane of the next LFNN layer. We add additional discussion in the caption of Method Figure 4 to clarify it.
- Neuron A and D are paired neurons. B and C are another pair of neurons, which has no direct connection with A and D. We draw B and C to make the idea of rotation clearer because only drawing two neurons (A and D) might confuse the rotation with reflection.
- The X-activation is not the critical claim that should appear in the main body of this paper but a convenient approach to overcome a disadvantage of incoherent optical systems, i.e., lack of negative sign. It brings a 5% improvement in the three-layer CIFAR10 classification. We have to clarify that a 5% improvement in classification is not a small effect for deep learning. Anyway, it is not the core of this paper but one of many optional technical choices, such as the sparsification approach (Method Sec 2.1) and the trainable probability distribution (Sec S4).
- There are a lot of challenges to realizing practical ONNs. This paper reports a lot of designs to train non-differentiable neurons as an ONNs, which are important results as pointed out by Reviewer 1. Realizing nonlinear activation of optics is another challenging task but orthogonal to this paper. Anyway, there are different ways to realize X-activation. For example, one simple alternative is to leverage lens to invert the image, then apply graphene saturable absorbers to perform optical subtraction.

8. I suggest providing a supplemental document such as a multimedia file to show your hardware architecture, experiment processes and results.

- We add video to show the hardware architecture and how to conduct experiment using a computer. The captured images are reported in Section S4.

Reviewer #3 (Remarks to the Author):

This paper proposes a function learning method to train the optical neural network without strict physical modeling of the system. An electronic convolution neural network model is employed for modeling the optical neural network. During the training, parameters of the electronic and optical neural networks are updated iteratively for the convergence of the system modeling and task learning. The idea seems fancy, however, I have strong concerns about the efficacy, efficiency, and scalability of the proposed method. Detailed

concerns are:

1. (efficacy) Details of ablation analysis. The only comparison counterpart is ref. 9 of the main text. As stated in the main (line 167) and supplement (line 133), the proposed method can improve the testing accuracy on the MNIST test set from 19.72% to 90%+. However, few details are provided about the implementation of the comparison. If there is a revision process, I suggest the authors add details about the comparison process, especially the “point spreading functions”, the “analytical model” (line 130 supplement), and the input and output examples along with expectations in the supplement.

- Because of the pixel-wise inconsistency of the LC panels (Sec. S2.2), it is hard to finely calibrate the hardware to build an accurate analytical model. Instead, we measure the ‘point spreading function’ of each pixel to build the ‘analytical model’, which can include the imperfections of the hardware, e.g., noise, misalignment, and narrow viewing angles, in the forward model. Therefore, the ‘analytical model’ and the ‘point spreading functions’ are referred to the same method in our paper. We add more discussion about the comparisons in the main manuscript and the supplementary document (Sec. S3.3).

2. (efficacy) Other comparison recommendations. There are many training methods without accurate interrogation of physical parameters for optical neural networks, which I will recommend for proof of the efficacy of the proposed method. Some examples are gradient calculation through finite-difference (ref. 6 main texts) and genetic algorithm (<https://doi.org/10.1021/acsphotonics.1c00035>).

- We compare the suggested methods and report the results in the main manuscript (Tab. 1) and more details in the supplementary document (Sec. S3.3).

3. (efficiency) In the supplement, the training of a neural network with the proposed method takes as many as thousands of epochs.

Since each epoch takes about 4 minutes, it would take several days to train the neural networks in this work, which is quite long considering the scale of the network. I suggest the authors add some discussions on the efficiency of the proposed training method.

- We add discussion of improving training speed in the end of the paper.

4. (scalability) In this work, the network scale are limited to four-layer amplitude-only modulation, and the capability is limited to binary classification, both of which are not close to real-world tasks. I suggest the authors add numerical exploration on more advanced tasks like 10-class classification with a larger number of layers, which is actually still quite easy tasks compared with an electronic neural network.

- Besides two binary classification tasks, our experiments include two 10-class classifications on MNIST and CIFAR10. MNIST is for the classification of hand-written

digits (Figure 7, Sec S4.1 and Sec S4.2). CIFAR10 is for the classification of real-world objects (Figure 7 and Sec S4.3). Both of them are commonly used classification tasks. In addition, we conduct a depth estimation task (Figure 9 and Sec S4.6), which has not been implemented by ONN. Compared with previous ONN researches, such as Ref 4-6, the complexity of our tasks is comparable or even higher.

- We are highly interested in training more layers of ONNs, e.g., 6 to 10 layers. However, as discussed near the end of the main manuscript, we will first improve the training speed to make the dense-layer ONN experiments more efficient.

5. (scalability). As an optical computing method in free space, I think they have a similar major problem to other optical neuromorphic computing methods. How do the authors plan to scale their system (which uses conventional and mature optical components) to a form factor and performance which is relevant for the field of AI and machine learning? The computing density is still very limited due to the liquid crystal used in this method. The current computing performance, as described in my previous questions, seems very limited compared with other optical computing methods or electronic methods.

- In this work, the proposed FL paradigm trains a simple LFNN device that is supposed to have low computing power but achieves competitive ONN inference results compared with other ONN approaches like Ref 4-6, showing the potential of the FL paradigms. We have many plans to increase the computing power of the hardware. The proposed FL paradigm can train a large number of non-differentiable and simple neurons without strict calibration, making it a promising approach for the realization of scalable ONNs. For example, the current LFNN device has 3096 input and output neurons, respectively, resulting in 6M parallel neuron connections. For the next step, replacing the LC panels and light sources with 4K panels will produce 64T parallel neuron connections. Integrating with high-speed light sources and detector (MHZ or THZ) yields E to Z per second theoretical computational power. In addition, we are highly interested in replacing LC with multiple SLM layers to modulate phase to further enhance the computing density for coherent ONN.

6. (application) Incoherent optical computing is a very exciting and challenging area, as it's very hard to model the light propagation with a large density. As the authors claim incoherent optical neural network, I would expect they show practical applications on normal scenes, e.g. object recognition in real-world, or depth estimation for a 3D sample. I think such an experiment will significantly improve the impact of this work.

- As a proof of concept, we train and test the LFNN in a closed environment using incoherent light sources to verify the possibility of the direct inference of real-world objects' signals. However, it is systematic work that requires a lot of additional efforts, such as collecting datasets. Thank you for your advice and we are seriously considering it as our next step.

Appendix:

```
import torch
import torch.nn as nn
import torchvision
import torchvision.transforms as transforms

# Check Device configuration
device = torch.device('cuda' if torch.cuda.is_available() else 'cpu')

target = 'MNIST_1L'
# target = 'MNIST_2L'
# target = 'CIFAR10_3L'

# Define Hyper-parameters
if target is 'MNIST_1L':
    input_size = 784
    hidden_size = 784
    num_classes = 10
    num_epochs = 50
    batch_size = 128
    learning_rate = 0.001
    num_layer = 1
    ir = 28 * 28
    # MNIST dataset
    train_dataset = torchvision.datasets.MNIST(root='../data',
                                              train=True,
                                              transform=transforms.ToTensor(),
                                              download=True)

    test_dataset = torchvision.datasets.MNIST(root='../data',
                                              train=False,
                                              transform=transforms.ToTensor())

# Fully connected neural network
class NeuralNet(nn.Module):
    def __init__(self, input_size, hidden_size, num_classes):
        super(NeuralNet, self).__init__()
        self.fc1 = nn.Linear(input_size, num_classes)

    def forward(self, x):
        out = self.fc1(x)
```

```
return out
```

```
elif target is 'MNIST_2L':
```

```
    input_size = 784
```

```
    hidden_size = 784
```

```
    num_classes = 10
```

```
    num_epochs = 50
```

```
    batch_size = 128
```

```
    learning_rate = 0.001
```

```
    num_layer = 2
```

```
    ir = 28 * 28
```

```
    # MNIST dataset
```

```
    train_dataset = torchvision.datasets.MNIST(root='../data',  
                                              train=True,  
                                              transform=transforms.ToTensor(),  
                                              download=True)
```

```
    test_dataset = torchvision.datasets.MNIST(root='../data',  
                                             train=False,  
                                             transform=transforms.ToTensor())
```

```
    # Fully connected neural network
```

```
    class NeuralNet(nn.Module):
```

```
        def __init__(self, input_size, hidden_size, num_classes):
```

```
            super(NeuralNet, self).__init__()
```

```
            self.fc1 = nn.Linear(input_size, hidden_size)
```

```
            self.bn1 = nn.BatchNorm1d(hidden_size)
```

```
            self.relu = nn.ReLU()
```

```
            self.fc2 = nn.Linear(hidden_size, num_classes)
```

```
        def forward(self, x):
```

```
            out = self.fc1(x)
```

```
            out = self.bn1(out)
```

```
            out = self.relu(out)
```

```
            out = self.fc2(out)
```

```
            return out
```

```
elif target is 'CIFAR10_3L':
```

```
    input_size = 3072
```

```
    hidden_size = 3072
```

```
    num_classes = 10
```

```
    num_epochs = 50
```

```

model = NeuralNet(input_size, hidden_size, num_classes).to(device)

# Loss and optimizer
criterion = nn.CrossEntropyLoss()
optimizer = torch.optim.Adam(model.parameters(), lr=learning_rate)

# Train the model
total_step = len(train_loader)
for epoch in range(num_epochs):
    for i, (images, labels) in enumerate(train_loader):
        # Move tensors to the configured device
        images = images.reshape(-1, ir).to(device)
        labels = labels.to(device)

        # Forward pass
        outputs = model(images)
        loss = criterion(outputs, labels)

        # Backpropagation and optimization
        optimizer.zero_grad()
        loss.backward()
        optimizer.step()

        if (i + 1) % 100 == 0:
            print('Epoch [{} / {}], Step [{} / {}], Loss: {:.4f}'
                  .format(epoch + 1, num_epochs, i + 1, total_step, loss.item()))

# Test the model
# In the test phase, don't need to compute gradients (for memory efficiency)
with torch.no_grad():
    correct = 0
    total = 0
    for images, labels in test_loader:
        images = images.reshape(-1, ir).to(device)
        labels = labels.to(device)
        outputs = model(images)
        _, predicted = torch.max(outputs.data, 1)
        total += labels.size(0)
        correct += (predicted == labels).sum().item()

```

```
print('Accuracy of the network on the 10000 test images: {}'.format(100 * correct / total))
```

REVIEWER COMMENTS

Reviewer #1 (Remarks to the Author):

The authors have sufficiently addressed the referee comments.

Reviewer #2 (Remarks to the Author):

In this paper, an optical neural network based on loose neuron array is proposed and a function learning method is presented. The feature of this structure is that loose neurons are used to realize programmable optical neural network to directly process incoherent light and RGB image related recognition and depth estimation tasks.

First of all, I personally think the author has done a good job overall. He used the electronic convolutional neural network to model the optical neural network. Optical neural network is an important topic in the field of intelligent optical computing, and it is very important to design functional or universal optical computing chips. According to my review experience in these years, the current mainstream training datasets of optical neural network are still MNIST series. The simple data set and low complexity of the network make it difficult to prove the application value of optical neural network in real scenes. The challenge lies in that the system error is difficult to correct. The size of the optical computing system error is positively correlated with the complexity of the system. Therefore, error correction algorithm is very important for building large-scale intelligent optical computing system, but there is still no universal error correction method. In addition, it is difficult to reconstruct the system. The structure of the existing optical neural network is difficult to reconstruct, so the calculation function is single. The programming of network parameters relies on relatively complex optical effects, and it is still difficult to write large-scale parameters quickly and accurately.

Now let's discuss the value of the work done by the author. The author uses incoherent light to design and implement optical field neural network, which belongs to free space optics. The author gives a detailed explanation of the network principle, but there are the following problems:

(1) Error analysis and correction of optical system. In this paper, from the recognition point of view, the performance of the network deteriorates dramatically when the CIFAR10 data set is adopted. The author does not analyze the reasons for the existence of this phenomenon, but simply mentions the influence of noise. So, for this kind of work, in a practical application scenario, error analysis and correction are very critical. As a reviewer/reader, I hope to see a comprehensive work that can solve practical problems in Nature Communications. Author can refer to T. Zhou, X. Lin, J. Wu, et al. Large-

scale neuromorphic optoelectronic computing with a reconfigurable diffractive processing unit[J]. Nature Photonics, 2021, 15: 367-373

(2) Latency and power analysis of network system. In the paper, the authors point out that this approach frees hardware from the burden of manual design, rigorous manufacturing and accurate assembly. However, judging from the programmability and performance of CIFAR10, this method is not even as good as traditional digital neural networks which rely on computer training. Furthermore, considering the response time brought by the adoption of liquid crystal devices, the authors do not discuss the time-delay and power-consumption correlation analysis of such an optical field neural network, after all, a meaningful work should provide a way to guide the task of solving real scenarios.

(3) Network performance. The computational performance of this method is limited, especially for CIFAR10. The author should try to improve the network performance.

(4) All-optical nonlinear activation. No matter in digital neural network or optical neural network, nonlinear activation improves the reasoning ability of the network. In such an optronic neural network proposed in this paper, there is no means to realize all-optical nonlinear activation, and the author uses digital nonlinear activation. Strictly speaking, such work is not called optical neural network, but optronic neural network. The author uses convolutional neural networks to guide the design of light field neural networks. According to the literature "Z. Gu, Y. Gao, X. Liu. Optronic convolutional neural networks of multi-layers with different functions executed in optics for image classification[J]. Optics Express, 2021, 29(4): 5877-5889" to further improve network design. In the above literature, convolutional neural networks can already be implemented in free space. In addition, the application of nonlinear activation functions in "Y. Zuo, B. Li, Y. Zhao, et al. All-optical neural network with nonlinear activation functions[J]. Optica, 2019, 6(9):1132-1137" is mentioned.

To sum up, I think the author should further improve and modify this work. The author should clearly understand that if the work is to be published in Nature Communications, it should have complete experiments, clear principles, detailed discussions sufficient to support the work, and potential application value. This will attract readers and provide an important complement to intelligent optical computing.

If not clear, the PDF file contains detailed reviewer comments.

Reviewer #3 (Remarks to the Author):

The authors have added many texts to address my concerns with better clarifications of this method. This idea is interesting and the field is a hot topic. However, the main problem remaining is the killer application of incoherent optical computing, which is the main claim of this paper. Because incoherent optical computing with modeling of the PSF with one or two layers has already been presented in previous methods for simple classifications on the optical bench. The authors leave this question to future work, while I still think the impact of this work will be significantly reduced without an experimental demonstration of incoherent optical computing in practical applications. Nevertheless, it's

a nice paper for proof-of-concept demonstrations with several steps forward in implementations of incoherent optical neural network.

Response Letter

Dear Reviewers,

We thank all the reviewers for their constructive feedback and valuable comments. In this revision, we take effort to address remaining concerns of reviewers. The primary modifications include:

- Section S2.5 in the supplementary document to report the system computing performance.
- Section S4.3.1 in the supplementary document to clarify possible reasons of the reduced prediction accuracy in the CIFAR10 dataset.

In the following, we response to each reviewer point-by-point.

Reviewer #1:

The authors have sufficiently addressed the referee comments.

A: Thank you for your comment.

Reviewer #2:

(1) Error analysis and correction of optical system. In this paper, from the recognition point of view, the performance of the network deteriorates dramatically when the CIFAR10 data set is adopted. The author does not analyze the reasons for the existence of this phenomenon, but simply mentions the influence of noise. So, for this kind of work, in a practical application scenario, error analysis and correction are very critical. Author can refer to T. Zhou, X. Lin, J. Wu, et al. Large-scale neuromorphic optoelectronic computing with a reconfigurable diffractive processing unit[J]. Nature Photonics, 2021, 15: 367-373

A: In our work, we assume the optical system contains a large number of unknown physical parameters without calibration, and seek to use the proposed functional learning paradigm to train such a *modeless* system. Therefore, we politely claim that there is no target optics for calculating the correction of our system. Instead, we report the optical characterization in Section S2.

Regarding the network deterioration in the CIFAR10 dataset, it is reasonable to expect the prediction accuracy of optical neural networks (ONNs) to behave lower than that of digital neural networks. This is mainly because the digital signal processing has theoretically infinite signal-to-noise ratio, unbounded computational complexity, and complete computational operations. Therefore, the performance of digital DNN can be seen as the upper limit of ONNs. In addition, as shown in Table xx, the performance of digital DNN also drops dramatically in the MNIST dataset, where our result has already achieved the metric close to this upper limit of digital DNN. Notably, the performance of (Chang, Julie, et al. "Hybrid optical-electronic ...", 2018) shares a similar dramatic drop in the CIFAR10 dataset (44.4%) and is in fact lower than ours. As such, we kindly argue that the performance reduction in complex dataset is not unique to our system but a common challenge to date.

There are several factors that can impact the neural network performance. First of all, the hardware-related network complexity and algorithm-related training complexity are partially attributed to this reduction. We note that the increased problem complexity in the CIFAR10 dataset is the primary reason, which almost all deep learning-based and most traditional algorithms suffer from. Notably, this drop is not unique to our algorithm or ONNs, but appears when using DNNs as well. The prediction accuracy of the conventional digital DNN drops dramatically from over 98% to about 50%, which is on par with our optical neural network implementation, confirming that the problem complexity is the primary reason. In summary, we have added S4.3.1 in the supplementary document to further explain possible reasons of performance reduction in the CIFAR10 dataset.

We kindly remind that even the most advanced DNNs face many unsolved problems. For example, the best mean IoU of digital neural networks is less than 50% in the DADA-seg dataset, while surprisingly in real-world applications their performance is often acceptable. Similarly, even though our system's performance in the CIFAR10 dataset is lower than 60%, results of hand-written digital classification, real-world object identification, and depth estimation are promising, showing its potential use in real-world applications.

To our best knowledge, up until the submission, our prediction accuracy in the CIFAR10 dataset is the highest among all ONNs. For such a relatively simpler dataset, i.e., MNIST, our performance is also on par with state-of-the-art papers on ONNs, while we are actually facing a more challenging incoherent system built

upon off-the-shelf components without fine-tuning. With the performance exhibiting similar to that of pure digital DNNs, i.e., about 50%, we envision optical neural networks can solve more challenging datasets with more advanced training algorithms incorporating CNNs.

(2) Latency and power analysis of network system. In the paper, the authors point out that this approach frees hardware from the burden of manual design, rigorous manufacturing and accurate assembly. However, judging from the programmability and performance of CIFAR10, this method is not even as good as traditional digital neural networks which rely on computer training. Furthermore, considering the response time brought by the adoption of liquid crystal devices, the authors do not discuss the time-delay and power-consumption correlation analysis of such an optical field neural network.

A: Thanks for your valuable comments. We have added an additional report in S2.5. Allow us kindly remind that our core contribution is the functional learning paradigm that can train the optical neural network with a large number of unknown physical parameters and uncalibrated physical neurons, rather than the hardware implementation itself. Training hardware with known parameters or calibrating for the imperfection of realization is a common problem in many real-world problems. As shown in Table 1, the new functional learning paradigm dramatically outperforms existing training paradigms, which is the primary scope of our work.

Notably, although the algorithm can much better train the hardware, the final performance is also bounded by physical constraints, such as the signal-to-noise ratio of the system. To this point, the concern of 'not even as good as traditional digital neural networks' doesn't apply only to our method, but a general case to almost all ONNs. This observation indicates that the theoretical and practical prediction accuracy of optical neural networks can be lower than digital neural networks because digital signal processing has theoretically infinite signal-to-noise ratio, unbounded computational complexity, and complete computational operations. On the other side, the optical neural network is a promising alternative with a high bandwidth.

For LC panels that are used to modulate the attenuation field, possible negative effects due to the response time of LC cells can be negligible, since the parameters of LC planes are mostly static during the inference. We would also like to kindly note that we have built our low-cost prototype using 60Hz LC panels to showcase the performance lower bound of our design paradigm, while commercial panels with 120Hz or even 180Hz are already on the market that when adopted, it is reasonable to expect a much better performance. In addition, with respect to optional light sources, there is no principle difficulties to apply laser or micro-LED to achieve much faster response times.

Please allow us to further comment on the power-consumption concern. If we understand the reviewer correctly, the concern was raised due to the possible light efficiency loss of light rays passing through LC layers and the required power consumption of driving LC cells. To this point, we kindly agree that the use of state-of-the-art LC panels may suffer from these problems. However, it would be non-trivial effort for us in accurately characterizing one specific LC panel model, which is also orthogonal to the scope of the proposed functional learning scheme. We kindly note that the programmable advantages and the low-cost of devices actually buy us credits.

(3) Network performance. The computational performance of this method is limited, especially for CIFAR10.

A: We have added a thorough analysis of the performance in the CIFAR10 dataset. In this work, we have proposed many novel designs to improve performance, including a training paradigm, an activation layer, a trainable classification layer, and so on. These efforts have dramatically improved the performance from about 10% to the current scores. The current performance in the CIFAR10 dataset is already close to that of digital DNNs, which can be viewed as an upper limit of optical neural networks in equal conditions. Up to now, this is also the best performance of the optical neural network in the CIFAR10 dataset, let alone it is achieved in an incoherent optical system.

For further improvement, we believe introducing the idea of CNN into the training process would be the key to enhancing the performance of optical neural networks in a complex dataset containing real-world objects. Yet, it is a challenging task that might need years to solve. Note that existing CNN algorithms cannot be directly adapted to our case because of the paradox between the spatial invariance requirement of CNNs and the spatial inconsistency of the hardware neurons. Detailed discussion can be found in Section S4.3.4.

In summary, we have achieved a concrete result that dramatically improves the performance of optical DNN to a level close to its upper limit. Next, we are highly motivated to devote ourselves into the research of CNN-based functional learning. We would highly appreciate staged acknowledgement to explore follow-up chances and funding for the following research.

(4) All-optical nonlinear activation. No matter in digital neural network or optical neural network, nonlinear activation improves the reasoning ability of the network. In such a optronic neural network proposed in this paper, there is no means to realize all-optical nonlinear activation, and the author uses digital nonlinear activation. Strictly speaking, such work is not called optical neural network, but optronic neural network. The author uses

convolutional neural networks to guide the design of light field neural networks. According to the literature “Z. Gu, Y. Gao, X. Liu. Optronic convolutional neural networks of multi-layers with different functions executed in optics for image classification[J]. Optics Express, 2021, 29(4): 5877-5889” to further improve network design. In the above literature, convolutional neural networks can already be implemented in free space. In addition, the application of nonlinear activation functions in “Y. Zuo, B. Li, Y. Zhao, et al.

A: We agree that all-optical nonlinear activation is not necessarily a critical part of a practical solution. A previous article called the mixed neural network hybrid optical-electronic neural networks (Chang, Julie, et al. “Hybrid optical-electronic convolutional neural networks with optimized diffractive optics for image classification.” Scientific reports). We do not use specific terms like ‘all-optics’, ‘optical-electronic’, or ‘optronic’ in our paper but the general term ‘optical neural network’ because our core contribution is the training paradigm for light modulation, and its application is not limited to specific nonlinear activation layers.

In our work, the design of light field neural networks is not guided by convolutional neural networks. We use CNN layers to fit the optics of the LFNN system in the Z-learning section, but the P-variable, i.e., the light modulation parameters, is not trained by CNN. Because the CNN requires shared weight between neurons, the actual hardware has inconsistency between neurons that requires neuron-wise optimal weights. A new algorithm is needed to address this paradox.

Some previous works have already realized free-space CNNs in free space ([Chang et al. 2018] and [Gu et al. 2021]), but these works have several fundamental differences from ours. First, we are trying to train challenging incoherent optical neural networks. Second, we are training a modelless hardware system with much fewer constraints. Third, we incorporate the modeling, calibration, and training of the hardware together, producing an end-to-end training paradigm that automatically adapts to the imperfection of hardware. Without the collaborated end-to-end modeling, calibration, and training algorithm, the performance of [Chang et al. 2018] and [Gu et al. 2021] is not clearly higher than ours. For example, [Chang et al. 2018] achieves at most 92.74% and 44.4% predict accuracy in the MNIST dataset and CIFAR10 dataset, respectively. [Gu et al. 2021] achieves 96.40% in the MNIST dataset (one CNN layer plus one FC layer), just slightly higher than our results (94.77% for two FC layers). In addition, we need to highlight that this work is concurrent with our research and not published during our submission.

In conclusion, we present state-of-the-art results in incoherent optical neural networks. Introducing the CNN into our functional learning paradigm is a promising direction, and we are highly motivated to continue the research. However, it is neither an intuitive task that could be done in a short notice nor the scope of this paper.

Reviewer #3:

The authors have added many texts to address my concerns with better clarifications of this method. This idea is interesting and the field is a hot topic. However, the main problem remaining is the killer application of incoherent optical computing, which is the main claim of this paper. Because incoherent optical computing with modeling of the PSF with one or two layers has already been presented in previous methods for simple classifications on the optical bench. The authors leave this question to future work, while I still think the impact of this work will be significantly reduced without an experimental demonstration of incoherent optical computing in practical applications. Nevertheless, it's a nice paper for proof-of-concept demonstrations with several steps forward in implementations of incoherent optical neural network.

A: Thanks so much for your comment. This paper contributes a new training paradigm for the precise training of optical neurons and an incoherent optical neural network with accuracy on par with coherent ones. Regarding the real-world inputs, even though this system aims for the proof-of-concept with the light sources in the lab, we also believe that directly feeding real-world objects' signals into the incoherent optical system would be an important milestone. Apparently, there are many unsolved challenges, such as how to bridge the domain gap between the light sources and real-world objects' signals. Due to the many practical issues like excessive workload and lack of funding, it is difficult for us to accomplish all trials in this paper. We are highly motivated to kick off the research, given this paper is out and receiving follow-up funding support.

REVIEWER COMMENTS

Reviewer #3 (Remarks to the Author):

The authors have addressed most of the concerns.

1. The illustration of figure 1 in the main manuscript requires referring to the supplementary information, which reduce the readability of the paper. Moreover, I suggest enhancing the writing for better consistency of the main manuscript.

2. I agree with reviewer 2 and have concerns about the details of the experimental configurations. As for the classification performance comparison with other works, I suggest the authors clearly describe their configuration of training and testing dataset in the manuscript, i.e., how many classes, how many samples per classes. Throughout the paper, the dataset configurations are not clear. As a result, all the comparison are not very valid. For example, the authors claim an improvement compare to (Chang, Julie, et al. "Hybrid optical-electronic ...", 2018) in the response. But this is not reasonable as there in the referenced paper, the number of classes tested on is 16, contrasted with 10 in the work submitted.

3. The authors should also discuss the source of noise and methods to reduce it, in the manuscript.

4. To address the concern on computing speed raised by reviewer 2, I think it is also necessary to clearly state the time it requires to train such neural networks and show its advantages over previous methods.

Response Letter

Dear Reviewer,

We are grateful for the timely and constructive feedback in the latest round of revision. In order to address the raised concerns, we make the following edits:

1. The illustration of figure 1 in the main manuscript requires referring to the supplementary information, which reduce the readability of the paper. Moreover, I suggest enhancing the writing for better consistency.

A: Thanks for your valuable comments. We have revised Figure 1 to improve its readability. The modification includes adding the related table slice to better assess the performance between various arrays and removing its connection to the supplementary document.

Note that due to the 5-pages limit of the main manuscript, we are unable to put all the details in it at this moment. We will be happy to take layout suggestions/comments from the editors later on and seek for adding more details in the main manuscript upon request.

2. I agree with reviewer 2 and have concerns about the details of the experimental configurations. As for the classification performance comparison with other works, I suggest the authors clearly describe their configuration of training and testing dataset in the manuscript, i.e., how many classes, how many samples per classes. Throughout the paper, the dataset configurations are not clear. As a result, all the comparison are not very valid. For example, the authors claim an improvement compare to (Chang, Julie, et al. "Hybrid optical-electronic ...", 2018) in the response. But this is not reasonable as there in the referenced paper, the number of classes tested on is 16, contrasted with 10 in the work submitted.

A: We have used the standard MNIST and CIFAR10 datasets. To clarify their configurations, we have added

a new 'Data' section in the end of the main manuscript under the 'Method' part.

We would like to note that in the 'Hybrid optical-electronic ...' paper, the 16-class dataset is the Google QuickDraw dataset, which contains drawn figures. The paper used the Google QuickDraw dataset in the simulation experiment. Alternatively, their and our papers all used the CIFAR10 (CIFAR-10) dataset that contains real-world objects to evaluate the actual optical output. The CIFAR10 dataset has only 10 classes (<https://www.cs.toronto.edu/~kriz/cifar.html>): airplane, automobile, bird, cat, deer, dog, frog, horse, ship, and truck.

3. The authors should discuss the source of noise and methods to reduce it, in the manuscript.

A: Thanks for the suggestion. We have added a new paragraph to discuss the source of noise and methods to reduce it between lines 351 to 366 of the main manuscript.

4. To address the concern on computing speed raised by reviewer 2, I think it is also necessary to clearly state the time it requires to train such neural networks and show its advantages over previous methods.

A: In the original version, we have discussed the training time in the 'Experiment' Section (line 262) of the supplementary document. To address this concern, in the revision, we have added a new discussion about the training speed in lines 335 to 344 of the main manuscript.

As shown in Table 1 and further discussed in the supplementary document, using equal training time, our method's predict accuracy can surpass existing methods over six times. Regarding the Forward Model, it is a pure simulated method and thus runs faster than all others. However, we would like to note that, in our investigation the Forward Model-based approach converges to a relative much lower accuracy, aka., 23.5. Although we have tried using a much larger number of training epochs based on the existing forward model, the performance would not show a noticeable improvement. As such, we kindly argue that ours significantly outperforms state-of-the-arts within the practical training time.